# Cation-driven phase transition and anion-enhanced kinetics for high energy efficiency zinc-interhalide complex batteries

Wei Zhong[1,2,3], Hao Cheng [1,2,3] ✉, Shichao Zhang[1], Laixi Li[1,2,3], Chaoqiang Tan[1,2], Wei Chen [4] & Yingying Lu [1,2,3] ✉

Aqueous Zn-halogen batteries, valued for high safety, large capacity, and low cost, suffer from the polyhalide shuttle effect and chaotic zinc electrodeposition, reducing energy efficiency and lifespan. Here we show a cation-driven positive electrode phase transition to suppress the shuttle effect and achieve uniform zinc electrodeposition, along with an anion kinetic enhancement strategy to improve energy efficiency and lifespan. Taking tetramethylammonium halide (TMAX, X = F, Cl, Br) as a subject, $TMA^+$ promotes oriented zinc (101) deposition on the negative electrode through electrostatic shielding, significantly extending cycling life. Concurrently, it captures $I_3^-$ on the positive electrode, forming a stable solid-phase interhalide complex that enhances coulombic efficiency. Compared to $I_3^-$ and $TMAI_3$, $X^-$ anions lower the Gibbs free energy differences of $I^- \to I_2X^-$ and $I_2X^- \to TMAI_2X$, accelerating $I^-/I_2X^-/TMAI_2X$ conversions and improving voltage efficiency. In TMAF-modified electrolytes, zinc interhalide complex batteries achieve a high energy efficiency of 95.2% at $0.2\,A\,g^{-1}$ with good reversibility, showing only 0.1% capacity decay per cycle over 1000 cycles. At $1\,A\,g^{-1}$, they show a low decay rate of 0.1‰ per cycle across 10,000 cycles. This study provides insights into enhancing energy efficiency and long-term stability for sustainable energy storage.

To address resource depletion and environmental pollution, large-scale deployment of renewable energy is essential, necessitating highly efficient storage systems. Among various options, rechargeable batteries are highly valued for their design flexibility, high efficiency, and ease of integration. Lithium-ion batteries stand out in the energy storage sector due to their reliable performance, long lifespan, and well-established technology[1]. However, the limited availability of lithium resources and safety concerns arising from toxic, flammable organic electrolytes restrict their broader application, highlighting the need for more sustainable and safer battery alternatives[2]. This has spurred research into sodium-ion and potassium-ion batteries, which leverage more abundant resources; yet, safety issues related to organic electrolytes remain a challenge[3,4]. Aqueous zinc-ion (Zn-ion) batteries have recently attracted attention as a promising alternative, offering advantages in safety, high theoretical capacity ($820\,mAh\,g^{-1}$ or $5855\,mAh\,cm^{-3}$), and low redox potential ($-0.762\,V$ vs. the standard hydrogen electrode, SHE)[5]. Various intercalation positive electrode materials, including metal oxides[6–8] and Prussian blue analogues[9], have been explored for compatibility with Zn negative electrodes. However, the high electron density of $Zn^{2+}$ leads to slow insertion/extraction kinetics, constraining the rate performance of these systems. Furthermore, the repeated insertion and extraction of $Zn^{2+}$ during cycling

[1]State Key Laboratory of Chemical Engineering, Institute of Pharmaceutical Engineering, College of Chemical and Biological Engineering, Zhejiang University, Hangzhou 310027, China. [2]ZJU-Hangzhou Global Scientific and Technological Innovation Center, Zhejiang University, Hangzhou 311215, China. [3]Institute of Wenzhou, Zhejiang University, Wenzhou 325006, China. [4]Department of Applied Chemistry, School of Chemistry and Materials Science, Hefei National Research Center for Physical Sciences at the Microscale, University of Science and Technology of China, Hefei, Anhui 230026, China.
✉e-mail: Bob_hao@zju.edu.cn; yingyinglu@zju.edu.cn

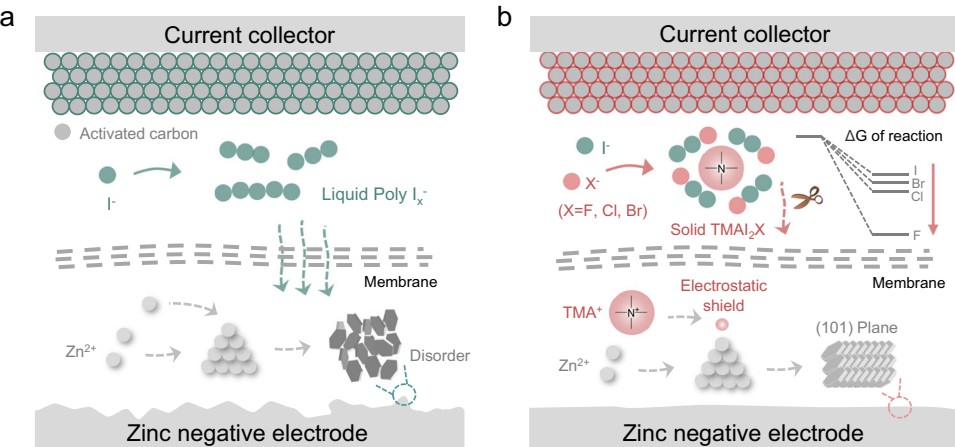

**Fig. 1 | Schematic illustration of cation-driven phase transition and anion-enhanced kinetics strategy. a** Polyhalide shuttle effect and chaotic Zn electrodeposition in conventional systems. **b** Cation-driven shuttle suppression and ordered Zn deposition coupled with anion-enhanced kinetics in modified systems.

can induce structural degradation of positive electrode materials, thereby reducing their cycle life.

Conversion-type positive electrodes, known for their rapid reaction kinetics, offer a promising approach to address these limitations. Among these, iodine positive electrodes are particularly attractive due to the abundant iodine reserves in nature (approximately 55 µg per liter in ocean water), high theoretical capacity (211 mAh g⁻¹ based on the I⁻/I₂ redox couple), and relatively high operating potential plateau (around 1.3 V vs. Zn/Zn²⁺)[10,11]. However, the strong interaction between $I_2$ and $I^-$ often leads to a liquid-liquid $I^-/I_3^-$ conversion during the cathodic $I^-/I^0$ redox reaction. The resulting highly water-soluble polyiodide ions can induce severe shuttle effects, causing notable capacity decay and low coulombic efficiency (CE), which in turn negatively impacts energy efficiency (EE)[12]. As we know, EE is a critical metric for battery energy storage, as high EE minimizes energy losses and operational costs. Additionally, polyiodide ions crossing the separator can directly interact with the Zn negative electrode, accelerating corrosion, dendrite formation, and shortening battery lifespan[13]. Therefore, effective inhibition of the polyiodide shuttle effect and achieving uniform Zn deposition are crucial for developing high-performance Zn-iodine batteries.

Various strategies have been explored to address these challenges, including designing positive electrode host materials to capture polyiodide ions[14–16], modifying separators to block their migration[17,18], and applying protective coatings on the negative electrode[19,20]. While these methods have shown some effectiveness, the preparation of iodine-based composite positive electrodes often requires the use of highly concentrated, volatile iodine solutions or vapors, which are toxic and pose environmental risks. Additionally, these methods typically involve complex, energy-intensive synthesis processes, leading to high costs and limiting their scalability. In contrast, electrolyte engineering offers a simpler, more scalable solution, with the potential to address issues at both the positive electrode and negative electrode, thus broadening its applicability. Previous approaches to reducing iodine dissolution have focused on minimizing free water content through high-concentration[21], eutectic[22], and gel electrolytes[23]. However, these solutions often suffer from high viscosity, low ionic conductivity, and high costs due to the need for expensive salts, which constrain their practical application. Recently, ligands capable of capturing polyiodide intermediates have emerged as promising alternatives, provided they meet the following criteria: (1) they contain cationic groups (e.g., quaternary ammonium ions with N+ centers or sulfonium ions with S+ centers) to ensure effective stabilization of polyiodides through electrostatic interactions; (2) they demonstrate good solubility and dispersibility in the electrolyte; and

(3) they are stable and resistant to decomposition[24,25]. Quaternary ammonium salts[26–30] and imidazolium salts[31], which are typical N+-containing compounds, have shown potential as polyhalide complexing agents, as they can effectively inhibit the shuttle effect by forming solid complexes through strong bonding with polyhalide ions. However, this liquid-solid conversion often exhibits slow reaction kinetics, impacting reversibility. Thus, there is an urgent need for an additive that not only captures polyiodide ions to suppress the shuttle effect but also enhances reaction kinetics to improve overall battery performance.

In this study, we employ a cation-driven phase transition approach to mitigate the shuttle effect and introduce an innovative anion-enhancement strategy to enhance reaction kinetics (Fig. 1a, b). Tetramethyl quaternary ammonium cation (TMA⁺, Supplementary Fig. 1), selected for its simple structure, serves as the primary additive component. TMA⁺ effectively immobilizes polyiodide ions, curbing the shuttle effect by forming solid-phase complexes due to its strong complexing capability. To investigate the impact on reaction kinetics, we used different halogen anions X⁻ (X = F, Cl, Br) as the anionic components of the additives. Experimental and theoretical analyses confirm that TMAX captures $I_3^-$, forming solid-phase interhalide complexes $TMAI_2X$. Compared to $I_3^-$ and $TMAI_3$, the introduction of X⁻ anions lowers the Gibbs free energy differences (ΔG) of $I^- \rightarrow I_2X^-$ and $I_2X^- \rightarrow TMAI_2X$, facilitating faster $I^-/I_2X^-/TMAI_2X$ conversions. Among the halogens, F shows notably enhanced performance. Additionally, TMA⁺ preferentially adsorbs onto the Zn negative electrode, creating a cationic electrostatic shield that promotes the formation of a Zn (101) crystal surface texture. This structural alignment enhances Zn deposition uniformity, effectively suppresses dendrite formation, and significantly extends the battery's cycle life. As a result, in the TMAF-modified electrolyte, the Zn-interhalide complex battery (ZICB) achieves a high average energy efficiency (AEE = 95.2%) and good reversibility, with minimal capacity decay of 0.1% per cycle over 1000 cycles at a low specific current of 0.2 A g⁻¹. At a higher specific current of 1 A g⁻¹, the ZICB demonstrates a low capacity decay of 0.1‰ per cycle across 10,000 cycles. This study introduces a pioneering anion enhancement strategy, advancing energy efficiency and reversibility toward more sustainable energy storage solutions.

## Results
### Association behavior between polyiodide ions and TMAX
To explore the association behavior between TMAX and polyiodide ions and to verify its effect in suppressing the shuttle effect, we introduced KI and TMAX separately into a KI₃ solution, observing any resulting precipitation. When KI was added to the dark orange I₃⁻

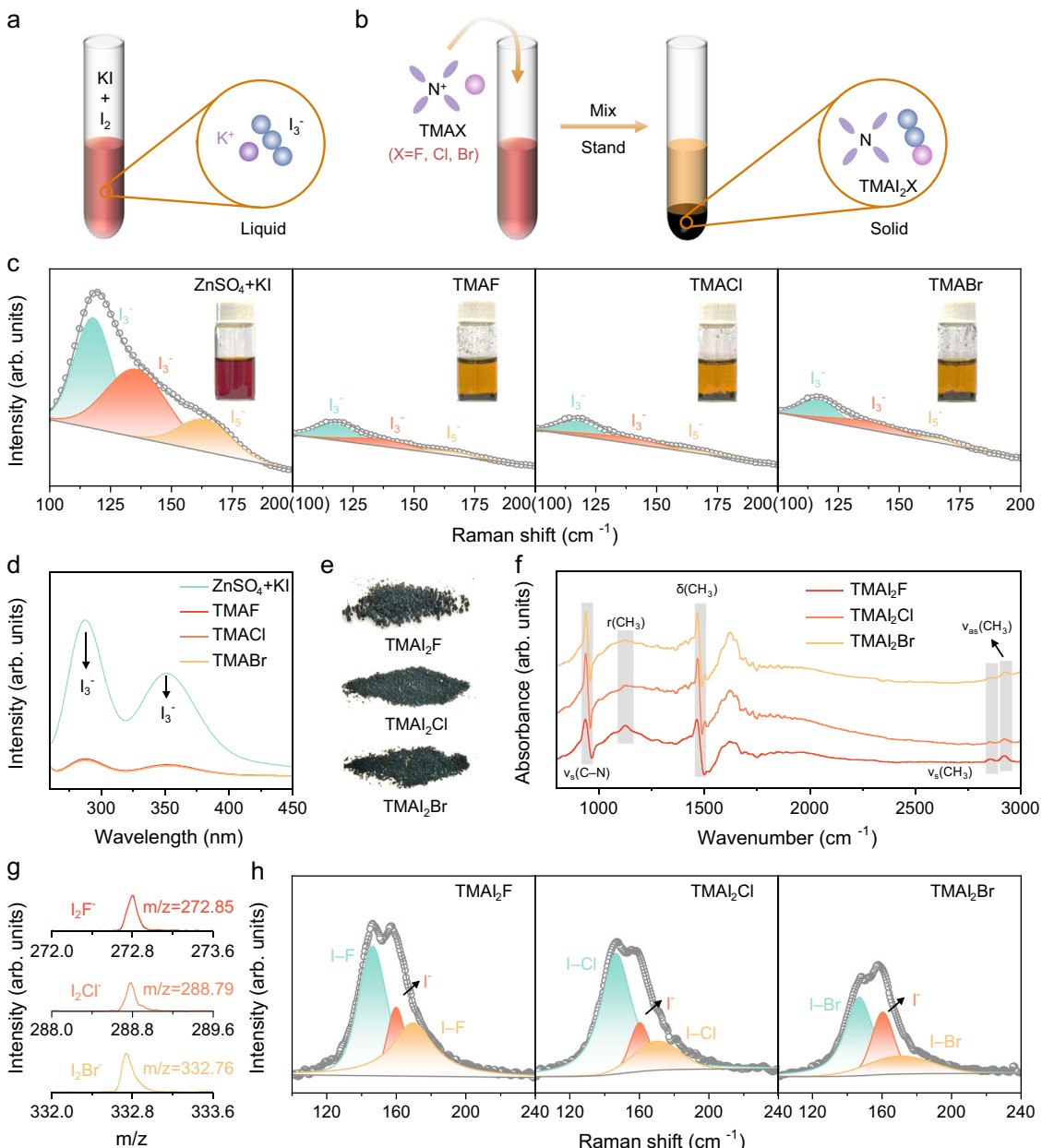

**Fig. 2 | Association behavior of TMAX in $I_3^-$ Solution. a, b** Schematic illustration of the association behavior of TMAX. **c** Fitted Raman spectra of the supernatant after mixing, inset images are photographs of the solutions after mixing. **d** UV-vis spectra of the supernatant. **e** Optical photos of solid precipitate powders. **f** FT-IR spectra, **g** Mass spectra and **h** Fitted Raman spectra of solid precipitate. Source data are provided as a Source data file.

solution, no significant color change was detected (Fig. 2a and Supplementary Figs. 2–4). However, upon adding TMAX to the $I_3^-$ solution, a distinct layering occurred, with a solid precipitate forming at the bottom of the vial and the supernatant becoming notably lighter in color (Fig. 2b, c). This suggests the formation of a solid complex between TMAX and $I_3^-$. To further confirm TMAX's association with $I_3^-$, we performed Raman and ultraviolet-visible spectroscopy (UV-Vis) on the supernatant (Fig. 2d). In the Raman spectrum (Fig. 2c), peaks at 118 and 136 cm$^{-1}$ correspond to $I_3^-$, while the peak at 165 cm$^{-1}$ is associated with $I_5^-$ [13,32,33]. $I_5^-$ ions are supposed to be formed from the disproportionation reaction: $2I_3^- \rightarrow I_5^- + I^-$ [34]. Notably, the signal intensities for both $I_3^-$ and $I_5^-$ decrease substantially after adding TMAX, indicating TMAX's strong binding effect on polyiodide ions. Supplementary Fig. 5 demonstrates that the polyiodide ion capture efficacy follows the order: TMAF > TMACl > TMABr. Additionally, the UV-Vis spectrum in Fig. 2d reveals a significant reduction in $I_3^-$ absorption peak intensities

at 288 nm and 350 nm upon TMAX addition [35], further confirming TMAX's ability to capture $I_3^-$ ions. We also examined the association process through visualization experiments. Supplementary Fig. 6 shows images captured after mixing $ZnSO_4 + KI$ and TMAX into $I_3^-$ solutions for 15 s. These images reveal that TMAX effectively captures polyiodide ions through a liquid-solid transformation, with the solid product formed by the reaction between TMAF and $I_3^-$ displaying more uniform size and distribution.

The solid precipitates formed after adding TMAX were isolated from the mixture for further analysis. Optical images revealed that the precipitates appeared as dark green powders (Fig. 2e). To determine the specific composition of these products, we conducted Fourier transform infrared (FT-IR) spectroscopy, mass spectrometry, and solid-state Raman spectroscopy analyses (Fig. 2f–h). In the FT-IR spectrum (Fig. 2f), the peaks at 939, 1126, 1464, 2852, and 2922 cm$^{-1}$ correspond to the symmetric stretching vibration of C−N ($v_s$(C−N)),

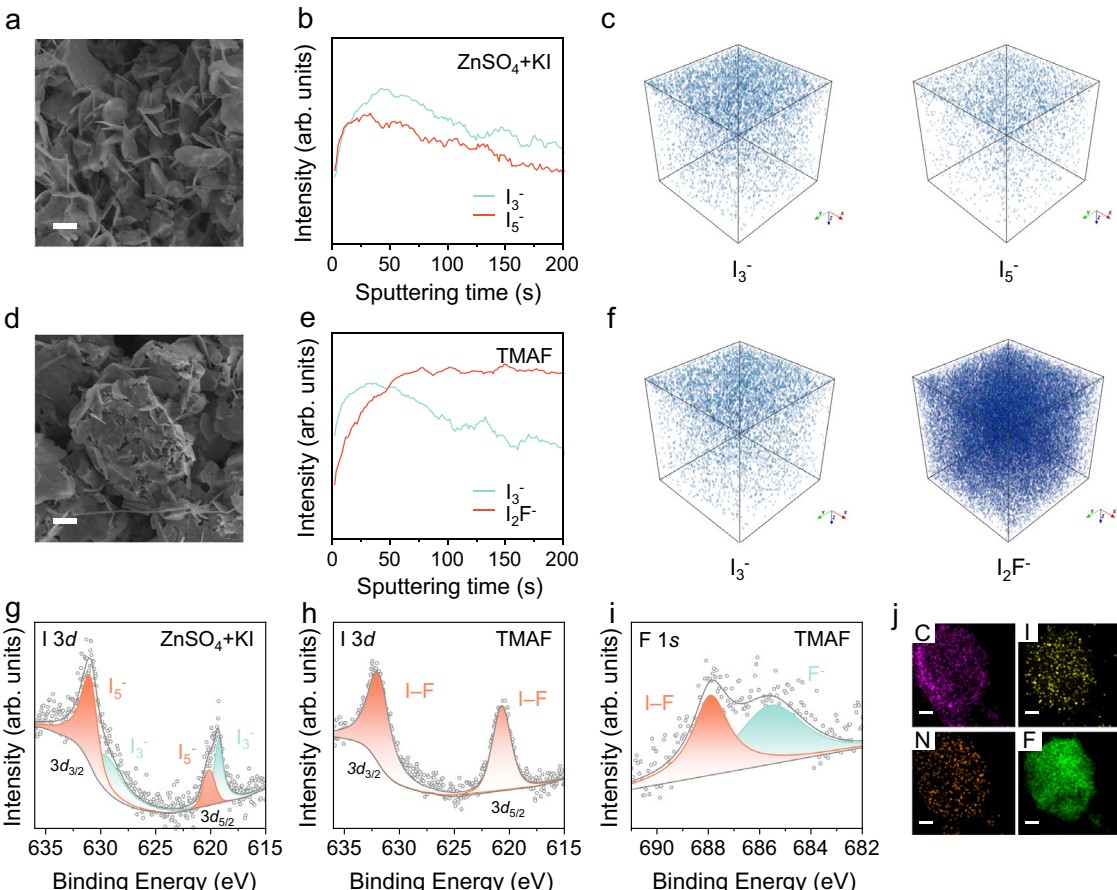

**Fig. 3 | Electrochemical Behavior of TMAX on the Positive Electrode.** SEM images of positive electrodes charged to 1.6 V at a specific current of 0.2 A g⁻¹ using **a** ZnSO₄ + KI and **d** TMAF electrolytes, scale bar=1 μm. ToF-SIMS negative in-depth ion profiles **b, e** and 3D visual maps **c, f** for positive electrodes charged to 1.6 V at a specific current of 0.2 A g⁻¹ performed in **b, c** ZnSO₄ + KI and **e, f** TMAF electrolytes.

Fitted I 3$d$ XPS spectra of positive electrodes charged to 1.6 V at a specific current of 0.2 A g⁻¹ using **g** ZnSO₄ + KI and **h** TMAF electrolytes. **i** Fitted F 1$s$ XPS spectra and **j** EDS element mapping of positive electrodes charged to 1.6 V at a specific current of 0.2 A g⁻¹ using TMAF electrolytes, scale bar=200 nm. Source data are provided as a Source data file.

the rocking vibration of CH₃ (r(CH₃)), the deformation vibration of CH₃ (δ(CH₃)), the symmetric stretching vibration of CH₃ (ν$_s$(CH₃)), and the asymmetric stretching vibration of CH₃ (ν$_{as}$(CH₃)), respectively, confirming the presence of TMA in the solid powder. The peak at 1620 cm⁻¹ is attributed to the deformation vibration of OH (δ(OH)), which may result from incomplete drying of the sample[36]. Mass spectrometry analysis revealed peaks with m/z values of 272.85, 288.79, and 332.76, which correspond to I₂F⁻, I₂Cl⁻, and I₂Br⁻, respectively (Fig. 2g). Solid-state Raman spectroscopy further confirmed the presence of I–F, I–Cl, and I–Br in the precipitate, with an additional I⁻ peak likely due to incomplete washing (Fig. 2h)[37,38]. These findings confirm that the composition of the solid product is TMAI₂X. The reactions for I₃⁻ formation and its capture by TMAX are illustrated in Eq. 1 and Eq. 2. TMAX interacts with I₃⁻ in the liquid phase, forming TMAI₂X via a liquid-solid phase transition, which immobilizes the polyiodide ions effectively.

$$I^- + I_2 \rightarrow I_3^- \tag{1}$$

$$I^- + 2I_2 + TMA^+ + X^- \rightarrow TMAI_2X + I_3^- \tag{2}$$

## Electrochemical behavior of TMAX on the positive electrode
After observing TMAX's significant interaction with I₃⁻ in solution, we further examined its electrochemical behavior within battery systems. Initially, both ZnSO₄ + KI and TMAX electrolytes appeared colorless

and transparent (Supplementary Fig. 7). Upon galvanostatic charging, a reaction occurred at the positive electrode-electrolyte interface in the ZnSO₄ + KI electrolyte, following the reaction 3I⁻ − 2e⁻ → I₃⁻, which was visibly marked by a dark orange solution in in situ optical microscopy images. In contrast, a notable lightening of color was observed in the TMAX electrolyte, indicating its substantial ability to decrease I₃⁻ concentration (Supplementary Fig. 8). To further investigate, we assembled a full battery and charged it to 1.6 V at a specific current of 0.2 A g⁻¹, subsequently analyzing the surface morphology of the activated carbon electrode using scanning electron microscopy (SEM). In both ZnSO₄ + KI and TMAX electrolytes, small flakes formed on the positive electrode surfaces. However, in the ZnSO₄ + KI electrolyte, these flakes were irregularly distributed across the electrode surface. By contrast, in the TMAX electrolyte, the flakes adhered more uniformly and tightly to the surfaces of the bulk activated carbon particles, suggesting a stronger affinity of TMAX for the activated carbon current collector (Fig. 3a, d and Supplementary Fig. 9).

To determine the precise composition of the small flakes on the electrode, a comprehensive analysis was performed utilizing time of flight secondary ion mass spectrometry (ToF-SIMS), X-ray photoelectron spectroscopy (XPS), and high-resolution transmission electron microscope (HRTEM). ToF-SIMS negative in-depth ion profiles reveal that in the ZnSO₄ + KI electrolyte only a thin layer of I₃⁻ and I₅⁻ is present on the electrode. After the sputtering time exceeds approximately 50 s, the content of both species decreases as sputtering time increases (Fig. 3b). The 3-dimensional (3D) ToF-SIMS ion distribution

maps also demonstrate the surface aggregation characteristic of $I_3^-$ (Fig. 3c). In contrast, in TMAF, a partial presence of $I_3^-$ is observed on the surface, but simultaneously, a significant amount of $I_2F^-$ emerges on the electrode. As sputtering time increases, the content of $I_2F^-$ gradually rises until it stabilizes, and the 3D ion distribution maps show a uniform distribution of $I_2F^-$ within the electrode (Fig. 3e, f). Similar patterns are also observed in TMACl and TMABr electrolytes, where $I_2Cl^-$ and $I_2Br^-$ are uniformly distributed on the electrode. However, with a slight difference, the contents of $I_2Cl^-$ and $I_2Br^-$ decline slightly during the later stages of sputtering (Supplementary Figs. 10–13). This indicates that among $I_2X^-$, $I_2F^-$ exhibits a more uniform distribution on the activated carbon electrode.

To further elucidate the state of I on the electrode surface, a detailed XPS analysis was conducted. For electrodes prepared in the $ZnSO_4$ + KI electrolyte, the I 3$d$ XPS spectra revealed two distinct pairs of peaks at 629.3 eV and 630.9 eV for the I 3$d_{3/2}$ orbital, and at 619.3 eV and 620.2 eV for the I 3$d_{5/2}$ orbital, corresponding to the $I_3^-$ and $I_5^-$ species, respectively (Fig. 3g). In the TMAF electrolyte, the I 3$d$ XPS spectrum showed two peaks at 620.7 eV and 631.9 eV, confirming the presence of I–F on the electrode. Additionally, the F 1$s$ XPS spectrum displayed peaks at 687.9 eV and 685.6 eV, which correspond to I–F and F$^-$, respectively (Fig. 3h, i). These findings collectively validate the presence of I–F on the electrode surface. Similarly, in TMACl and TMABr electrolytes, the I 3$d$ XPS spectra revealed peaks consistent with the presence of I–Cl and I–Br. Peaks at 201.2 eV in the Cl 2$p$ XPS spectrum and at 69.8 eV in the Br 3$d$ XPS spectrum further confirm the presence of I–Cl and I–Br, respectively (Supplementary Figs. 14, 15)[39–41]. HRTEM combined with energy-dispersive spectroscopy (EDS) mapping provided additional insights. For the positive electrode obtained in $ZnSO_4$ + KI electrolyte, only carbon (C) and I were detected, with C primarily attributed to the activated carbon substrate (Supplementary Fig. 16). Conversely, in the TMAX electrolyte, the positive electrode displayed the presence of C, N, I, and X elements, indicating the successful formation of $TMAI_2X$ interhalide complexes during the charging process (Fig. 3j and Supplementary Figs. 17–19). To investigate the distribution of the charging product $TMAI_2X$ on the activated carbon electrode, we conducted nitrogen (N2) adsorption experiments to characterize the Brunauer-Emmett-Teller specific surface area ($S_{BET}$) and pore volume of the activated carbon and employed focused ion beam-scanning electron microscope (FIB-SEM) to investigate the cross-section of a single activated carbon particle (Supplementary Figs. 20–22 and Supplementary Table 1). The decrease in $S_{BET}$ and pore volume values after charging and the enrichment of I and F on the surface of activated carbon indicate that the stable charging product $TMAI_2F$, formed in TMAF electrolytes, is distributed both on the surface and within the pores of the activated carbon, with the majority of the distribution occurring on the surface of the activated carbon. These comprehensive analyses confirm the formation and uniform distribution of interhalide complexes $TMAI_2X$ on the electrode, highlighting TMAX's role in stabilizing the electrode surface.

## Halide anion-enhanced positive electrode reaction kinetics

After successfully capturing $I_3^-$ through TMA$^+$-driven phase transition, halogen anions (X$^-$) were introduced to enhance the positive electrode reaction kinetics. The electrochemical performance of the electrolyte containing TMAX was examined using cyclic voltammetry (CV) (Fig. 4a, Supplementary Figs. 23a and 24a). Two distinct peaks near 1.19 V and 1.28 V were observed, corresponding to the redox processes of the $I^-/I_2X^-$ redox couple. As the scan rate increased from 1 mV s$^{-1}$ to 5 mV s$^{-1}$, the peak currents ($I_p$) showed a linear relationship with the square root of the scan rate (Fig. 4b, Supplementary Figs. 23b and 24b), indicating diffusion-controlled kinetics. Using the Randles-Sevcik equation[42] and the slope (k) of the fitted linear lines, the diffusion coefficients ($D_0$) of I$^-$ in different electrolytes were calculated, as detailed in Supplementary Note 1, with results summarized in

Supplementary Table 2. $D_0$ values during oxidation and reduction displayed the trend: TMAF>TMACl>TMABr. Specifically, in the TMAF electrolyte, $D_0$ was $2.59 \times 10^{-6}$ cm$^2$ s$^{-1}$ during oxidation and $3.12 \times 10^{-6}$ cm$^2$ s$^{-1}$ during reduction, indicating that TMAF significantly enhances electrochemical kinetics compared to other TMAX variants.

To further assess the impact of halide ions on reaction kinetics, linear scan voltammetry (LSV) was conducted to analyze the iodine oxidation reaction (IOR) and iodine reduction reaction (IRR). Tafel slopes, derived from LSV curves (Supplementary Fig. 25), reveal kinetic improvements with TMAF. The Tafel slope for IOR in TMAF (38.8 mV dec$^{-1}$) is lower than that in TMACl (43.6 mV dec$^{-1}$) and TMABr (49.1 mV dec$^{-1}$), indicating more favorable positive electrode kinetics in TMAF electrolyte (Fig. 4c). Similarly, for IRR, Tafel slopes followed the order TMABr (99.3 mV dec$^{-1}$), TMACl (80.5 mV dec$^{-1}$), and TMAF (48.1 mV dec$^{-1}$), further demonstrating that TMAF significantly enhances the kinetics of both IOR and IRR (Fig. 4d). These results indicate that TMAF effectively boosts the kinetic performance of both oxidation and reduction processes, establishing it as the most efficient halide anion for enhancing positive electrode reactions.

To elucidate ion diffusion phenomena during charging and discharging, ion diffusion coefficients were measured using the galvanostatic intermittent titration technique (GITT), with calculations outlined in Supplementary Note 2[43,44]. The GITT curves and corresponding diffusion coefficients are shown in Supplementary Fig. 26 and Fig. 4e. During charging, diffusion coefficients in the TMAF electrolyte remained within $10^{-6}$–$10^{-7}$ cm$^2$ s$^{-1}$, whereas in TMACl and TMABr ranged from $10^{-6}$–$10^{-8}$ cm$^2$ s$^{-1}$ and $10^{-6}$–$10^{-9}$ cm$^2$ s$^{-1}$, respectively. Notably, in TMABr, the diffusion coefficient experienced a significant decrease in the potential range of 1.3 to 1.6 V, suggesting slower reaction kinetics due to the transition from I$^-$ to $I_2Br^-$. In contrast, TMAF maintained a more stable diffusion coefficient, indicating enhanced kinetics between I$^-$ to $I_2F^-$. The stable range of diffusion coefficients during both charging and discharging in TMAF also points to reliable reversibility of the reaction.

To assess interfacial charge transfer kinetics, distribution relaxation time (DRT) was obtained through in situ electrochemical impedance spectroscopy (EIS) and deconvolution of the EIS data. The DRT spectrum (Fig. 4f) displayed two primary peaks, P1 and P2: P1 is attributed to electron transfer of redox species, while P2 is related to interfacial charge exchange and electrolyte mass transfer[45]. At charging potentials of 0.9 V, 1.1 V, and 1.5 V, P1 shifted toward lower timescales in TMAF, and the distribution function g(τ) of P2 decreased from TMABr to TMACl and further to TMAF. This indicates faster redox reactions and enhanced interfacial charge exchange kinetics in TMAF compared to the other electrolytes.

To explain the kinetic differences among halide ions, density functional theory (DFT) calculations were performed. The ΔG values for reactions of $I^- \to I_2X^- \to TMAI_2X$ (X = F, Cl, Br, I) were calculated (Fig. 4g, Supplementary Table 3 and Supplementary Data 1). The ΔG values for $I^- \to I_2X^-$ follow the trend $I_2F^- < I_2Cl^- < I_2Br^- < I_3^-$. Lower ΔG values indicate higher reaction spontaneity, with $I_2F^-$ being the most favorable and stable due to its lowest ΔG. This suggests that TMAF facilitates the acceleration of the reaction rate and enhances the iodide redox kinetics[46–48]. Structural analysis (Fig. 4h) revealed that all $I_2X^-$ species exhibit linear or near-linear trihalide structures[49]. Among them, $I_2F^-$ has the shortest I–X bond length, indicating the strongest binding affinity, which decreases in the order F > Cl > Br > I. Electrostatic potential (ESP) distributions (Fig. 4i, Supplementary Fig. 27 and Supplementary Data 2) show that iodine atoms in $I_3^-$ carry negative charges, while bonding with electronegative halogens (F, Cl, Br) causes electron transfer, resulting in the I atom adjacent to X$^-$ carrying a partial positive charge. F, with its high electronegativity, induces a relatively high positive charge on the adjacent I atom. These findings are consistent with XPS observations for the positive electrode (Fig. 3g–i and Supplementary Figs. 14, 15).

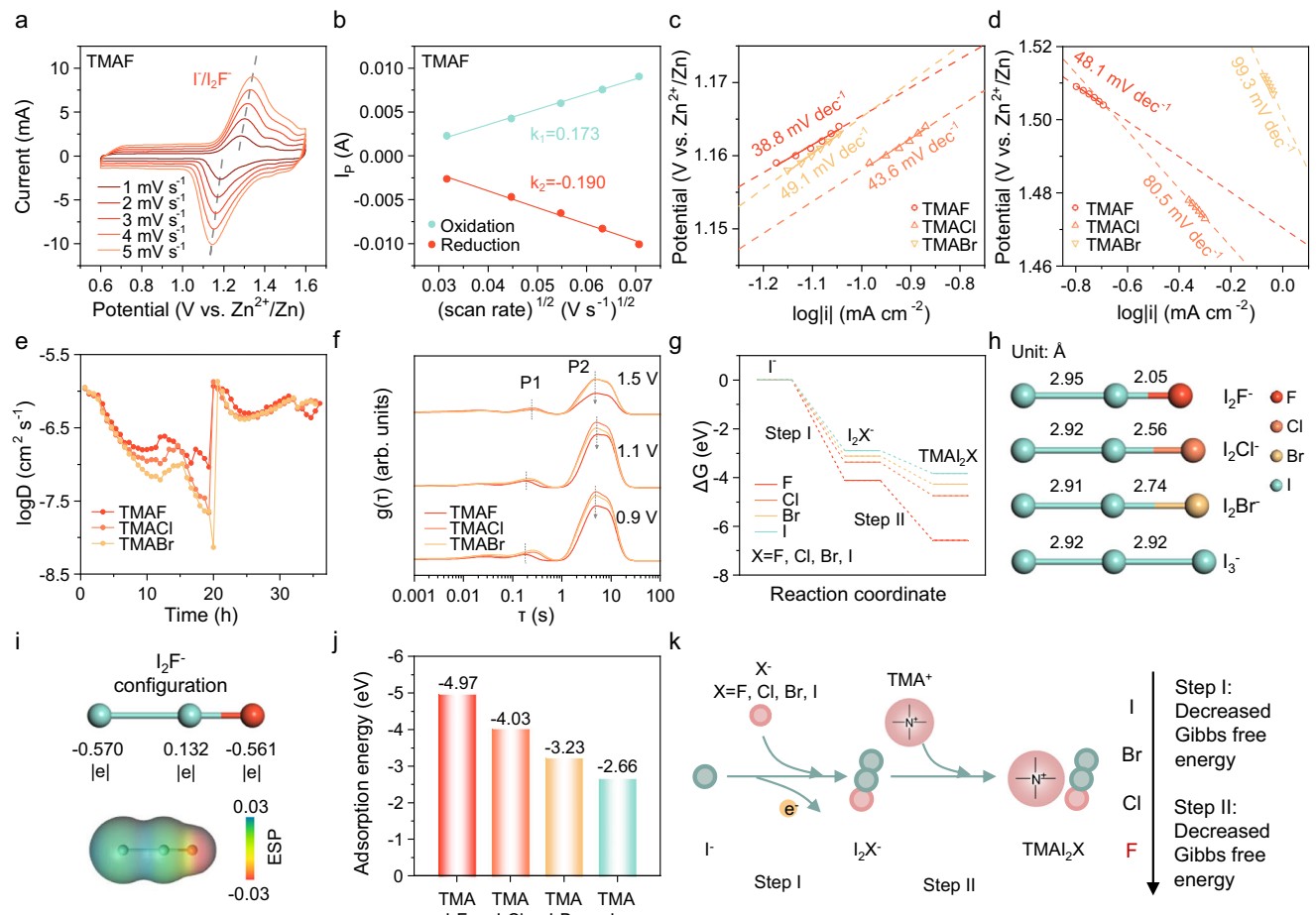

**Fig. 4 | Kinetic analysis of ZICBs using TMAX electrolyte. a** CV profiles of ZICBs using TMAF electrolyte. **b** Relationship between peak current and square root of scan rate. Tafel plots of **c** IOR and **d** IRR. **e** I⁻ diffusion coefficient in different electrolytes. **f** DRT curves charging to different potential. **g** The Gibbs free energy ladder diagram. **h** Configurational structure and bond length of $I_2X^-$. **i** Valence states (atomic Mullikan charge) and ESP of $I_2F^-$. **j** Adsorption energy $TMAI_2X$ on carbon layer. **k** Schematic illustration of the halide anion-enhanced behavior. Source data are provided as a Source data file.

The ΔG values for the reaction $I_2X^- \rightarrow TMAI_2X$ also follow the trend: $TMAI_2F < TMAI_2Cl < TMAI_2Br < TMAI_3$ (Fig. 4g). Lower ΔG values favor the reaction, indicating that $TMAI_2F$ is more easily formed and exhibits enhanced kinetic performance. We also calculated the adsorption energy of $TMAI_2X$ on the activated carbon electrode surface (Fig. 4j, Supplementary Figs. 28–31 and Supplementary Data 3). The adsorption energy of $TMAI_2X$ on the activated carbon electrode follows the trend: $TMAI_2F < TMAI_2Cl < TMAI_2Br < TMAI_3$. Among them, the adsorption energy of $TMAI_2F$ was the lowest, which indicated that it had the strongest adsorption force and was most likely to be attached to the surface of the activated carbon electrode. This stronger adsorption energy implies that $TMAI_2F$ will firmly adhere to the electrode surface and diffuse more rapidly, leading to a more uniform distribution. In contrast, the adsorption energies of the $TMAI_2Cl$ and $TMAI_2Br$ complexes are weaker, which may result in larger adsorption-desorption fluctuations, leading to unstable adsorption on the electrode surface and a more uneven distribution. This is consistent with experimental observations (Fig. 3c, f).

The computational results align with the aforementioned experimental findings, demonstrating that TMAF significantly enhances the reaction kinetics and ensures uniform distribution on the activated carbon electrode surface. The enhancement of the $I^- \rightarrow TMAI_2X$ reaction occurs through a two-step mechanism (Fig. 4k). In Step I (Eq. 3), I⁻ and X⁻ combine undergo electron loss to form $I_2X^-$, where X⁻ lowers the reaction's ΔG (with F having the most pronounced effect), making the redox process energetically favorable. In Step II

(Eq. 4), $TMA^+$ combines with $I_2X^-$ to form solid $TMAI_2X$. The $I_2F^- \rightarrow TMAI_2F$ reaction has the lowest ΔG, making it the most easily formed. Therefore, F chemistry plays a dual role in enhancing the entire reaction pathway, contributing to improved reaction kinetics.

$$2I^- - 2e^- + X^- \rightarrow I_2X^- \tag{3}$$

$$TMA^+ + I_2X^- \rightarrow TMAI_2X \tag{4}$$

## Mechanism of Zn dendrite suppression by TMA⁺

To assess the stability of Zn deposition, Zn||Zn symmetric cells were tested in $ZnSO_4$ + KI and TMAX electrolytes. The $ZnSO_4$ + KI cell short-circuited after 255 h at a current density of 4 mA cm⁻² and a capacity of 1 mAh cm⁻². In stark contrast, cells with TMAX electrolytes (X = F, Cl, Br) maintained stable Zn plating/stripping for over 1400 h—more than 5 times longer than with $ZnSO_4$ + KI (Fig. 5a and Supplementary Figs. 32–34). Zn||Cu asymmetric cells using the TMAF electrolyte under high areal capacity conditions (10 mA cm⁻² and 10 mAh cm⁻²) exhibited stable Zn plating/stripping for 212 cycles with a CE of 99.78% —over 6 times longer than with the $ZnSO_4$ + KI electrolyte (Supplementary Figs. 35, 36). This result unequivocally demonstrates the enhanced reversibility of the Zn negative electrode in the TMAF electrolyte system[50].

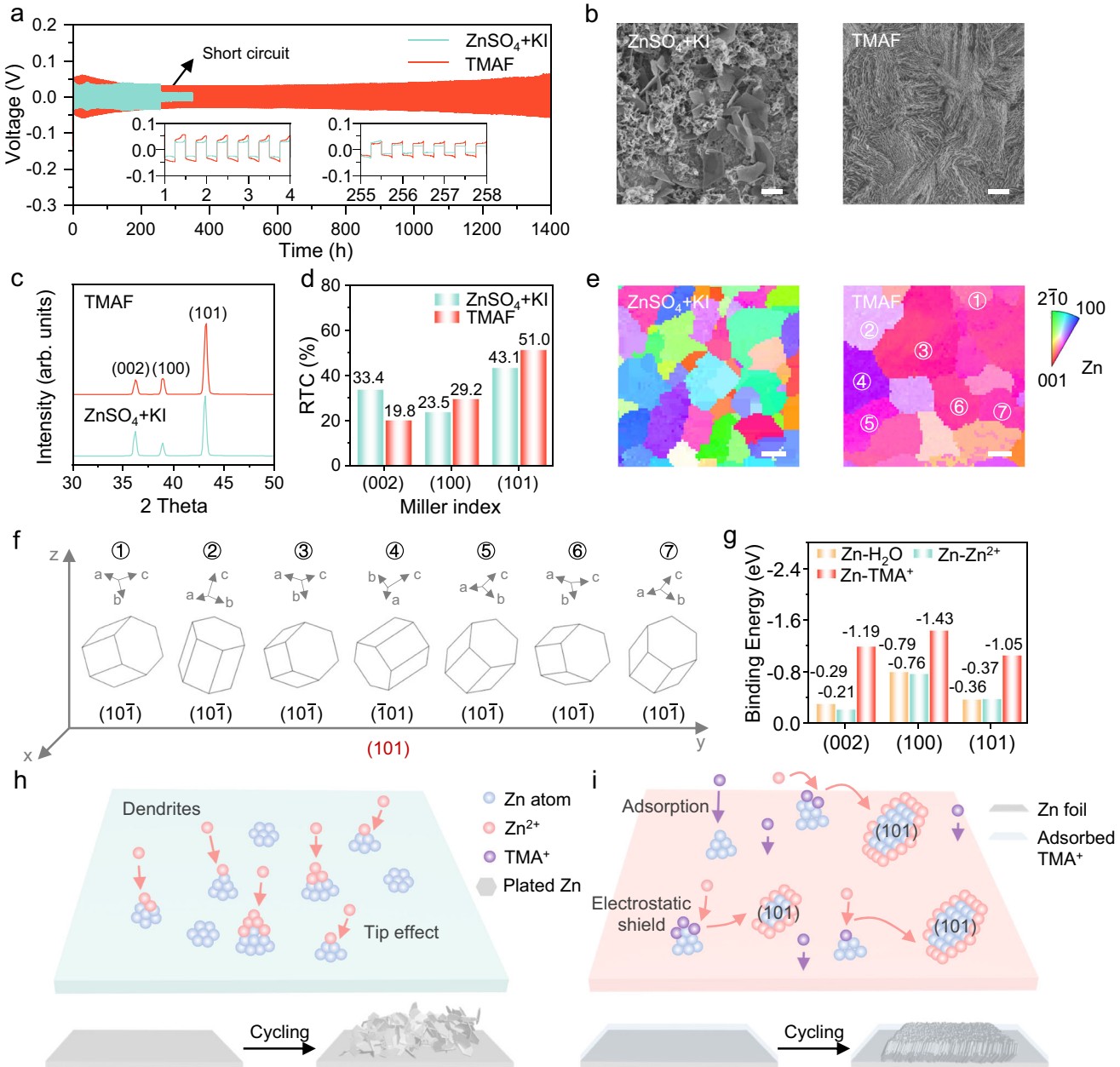

**Fig. 5 | Effect of TMAX on Zn deposition behavior. a** Galvanostatic Zn plating/stripping stability of symmetric Zn||Zn cells using $ZnSO_4$ + KI and TMAF electrolytes under 4 mA cm⁻², 1 mAh cm⁻². The inset curves provide enlarged views of 2-8 cycles and 510−516 cycles. **b** SEM images of Zn negative electrodes after 15 minutes of Zn deposition at 4 mA cm⁻² in $ZnSO_4$ + KI and TMAF electrolytes, scale bar=5 μm. **c** XRD patterns, **d** the corresponding RTC and **e** Inverse pole maps of Zn negative electrodes after 1 hour of Zn deposition at 4 mA cm⁻² in $ZnSO_4$ + KI and TMAF electrolytes, scale bar = 2 μm. **f** Crystal orientation analysis. **g** Binding energies of $H_2O$, $Zn^{2+}$ and TMA⁺ on Zn (002), Zn (100) and Zn (101) planes. Schematic illustration of **h** Zn dendrite formation and **i** electrostatic shielding effects. Source data are provided as a Source data file.

SEM images showed that Zn deposited in $ZnSO_4$ + KI was loose and dendritic, whereas Zn deposited in TMAX electrolytes was dense, homogeneous, and regularly arranged (Fig. 5b, Supplementary Figs. 37, 38). X-ray diffraction (XRD) analysis revealed a shift in Zn crystallographic orientations. Zn deposited in TMAX showed a decreased (002) plane intensity and increased (100) and (101) plane intensities compared to $ZnSO_4$ + KI (Fig. 5c). This change in orientation, quantified through the correlation coefficients (RTCs) of Zn facets[51], indicated a preference for the (101) orientation in TMAX, which is conducive to suppressing dendrite formation (Fig. 5d, Supplementary Figs. 39, 40, and Note 3). Additional evidence came from 2D grazing-incidence wide-angle X-ray scattering (2D-GIWAXS), where

Zn deposited in $ZnSO_4$ + KI showed random orientation, while in TMAX, the Zn (101) plane exhibited pronounced diffraction intensity, confirming the preferred (101) orientation (Supplementary Fig. 41). Electron backscatter diffraction (EBSD) analysis was employed to further elucidate the relationship between deposition morphology and crystallographic orientation. The inverse pole figure (IPF) map derived from EBSD reveals a colorful pattern on Zn deposited in $ZnSO_4$ + KI, indicating that the crystallographic planes are irregular and disordered. In contrast, the Zn deposited in TMAF exhibits a predominantly purple hue, all corresponding to the Zn (101) plane. The color variations among different regions (①-⑦) are attributed to the varying orientations of the (101) plane (Fig. 5e, f), which is

consistent with the differently oriented stacking directions of crystallographic planes observed in Fig. 5b. 3D confocal imaging further confirmed the compact Zn deposition in TMAX electrolytes (Supplementary Fig. 42). While Zn deposited in $ZnSO_4 + KI$ displayed a rough, non-uniform surface up to 51.1 μm thick, the Zn in TMAX was more uniform and compact, with an average thickness of around 20 μm and significantly reduced surface roughness ($S_a$), dropping to less than half of the original 6.64 μm. The nucleation overpotential test (Supplementary Fig. 43) revealed a larger nucleation overpotential in TMAX, promoting more uniform $Zn^{2+}$ deposition and mitigating dendrite formation, in agreement with SEM observations. To investigate the effect of TMAX on zinc deposition, chronoamperometry (CA) tests were conducted at a constant potential of −150 mV on Zn||Zn symmetric cells using $ZnSO_4 + KI$ and TMAF electrolytes (Supplementary Fig. 44). The results showed that in the TMAF electrolyte, Zn exhibited a consistent 3D diffusion process, maintaining a dense and flat surface morphology during the plating process. In contrast, in the $ZnSO_4 + KI$ electrolyte, Zn demonstrated a typical 2-dimensional (2D) diffusion mode with dendritic growth, which is consistent with the SEM results shown in Fig. 5b.

To understand how $TMA^+$ inhibits Zn dendrite formation, DFT calculations were conducted to compare the binding energies of $H_2O$, $Zn^{2+}$, and $TMA^+$ on Zn (002), Zn (100), and Zn (101) surfaces (Fig. 5g). Results showed that $TMA^+$ has lower binding energies on all three surfaces than $H_2O$ and $Zn^{2+}$, suggesting that $TMA^+$ preferentially adsorbs onto the Zn negative electrode. In addition, $TMA^+$ exhibits a lower adsorption energy on Zn (002) and Zn (100) planes compared to the Zn (101) plane, leading to its preferential adsorption on the Zn (002) and Zn (100) planes. Consequently, this exposure of the unconventional Zn (101) surface facilitates its growth, thereby achieving a Zn (101)-dominant texture[52]. The ordered and compact epitaxial growth of Zn along the (101) crystal plane enhances the stability of Zn negative electrode. Supplementary Fig. 45 illustrates this adsorption model, with differential charge density diagrams indicating that the N atom in $TMA^+$ strongly adsorbs onto the negative electrode (Supplementary Figs. 46-48). The Zn dendrite formation mechanism is depicted in Fig. 5h. During initial electrodeposition, $Zn^{2+}$ ions adsorb and nucleate unevenly on the rough electrode surface. As deposition progresses, $Zn^{2+}$ migrates toward initial nucleation points, leading to Zn protrusions that create focal points of electric field concentration due to the "tip effect". This effect causes $Zn^{2+}$ ions to accumulate at the protrusions, accelerating vertical growth and resulting in dendrite formation[53]. The electrostatic shielding effect of $TMA^+$ addresses this issue. Both experimental and computational results show that $TMA^+$ preferentially adsorbs onto the Zn surface, forming a shielding layer that reduces the "tip effect". This layer inhibits Zn dendrite formation and promotes uniform Zn deposition, with the Zn (101) plane emerging as the dominant crystalline facet (Fig. 5i).

To investigate the effect of TMAX on Zn corrosion, Tafel curve and LSV tests were conducted. The Tafel curves (Supplementary Fig. 49) show that the presence of TMAF reduced both the corrosion potential and current compared to the $ZnSO_4 + KI$ electrolyte, indicating that TMAF inhibited hydrogen evolution reaction (HER). Additionally, to better evaluate the inhibitory effect of TMAF on parasitic reactions, LSV tests were performed using a three-electrode configuration. A 2 M $Na_2SO_4$ solution was used to replace 2 M $ZnSO_4$ in order to eliminate the interference of $Zn^{2+}$ reduction on the HER measurements. The LSV results (Supplementary Fig. 50) show that the response current density of the $Na_2SO_4 + KI + TMAF$ electrolyte was significantly lower than that of the $Na_2SO_4 + KI$ electrolyte, indicating that TMAF effectively inhibited parasitic reactions. The inhibition of Zn corrosion by TMAF can be attributed to the specific adsorption of $TMA^+$ cations on the Zn surface, which induces uniform deposition and reduces the number of active sites on the surface.

## Electrochemical performance of ZICBs

Building on the understanding of TMAX's role in stabilizing both negative electrode and positive electrode, we evaluated the electrochemical performance of full ZICBs assembled with activated carbon positive electrodes and various electrolytes ($ZnSO_4 + KI$ and TMAX) in detail. Self-discharge behavior was assessed through an open-circuit voltage (OCV) test, where fully charged batteries were kept at OCV for 60 hours, followed by capacity recovery measurements. The resulting CE served as an indicator of self-discharge severity (Supplementary Fig. 51). Batteries with $ZnSO_4 + KI$ electrolyte displayed significant self-discharge, with CE dropping to 80.95%, whereas TMAX electrolytes effectively suppressed this effect, achieving a high CE of 98.32% with TMAF, demonstrating the strongest suppression of shuttle effects.

Figure 6a–c illustrate the CE, capacity, voltage efficiency (VE), and energy efficiency (EE) of ZICBs at 0.2 A g$^{-1}$. $ZnSO_4 + KI$ batteries suffered from rapid capacity decay and a fluctuating CE, failing after 194 cycles with only 66.5 mAh g$^{-1}$ capacity, reflecting a retention of just 59.1%. In stark contrast, batteries with TMAX electrolytes showed robust stability and higher capacities. Specifically, TMAF, TMACl, and TMABr maintained stable capacities of 155.0, 134.8, and 113.6 mAh g$^{-1}$, respectively, while TMAF showed a capacity decay rates as low as 0.1% per cycle over 1000 cycles. These results confirm that TMAX significantly enhances battery capacity and lifespan, with TMAF showing the most pronounced effect.

The average CE (ACE), VE (AVE), and EE (AEE) were calculated for $ZnSO_4 + KI$ and TMAX-based batteries (Supplementary Table 4, Fig. 6d). TMAX electrolytes substantially boosted these performance metrics, with results following the order TMAF > TMACl > TMABr. The improvement in CE is primarily due to TMAX's suppression of shuttle effects via liquid-solid phase transition, while the VE enhancement stems from TMAX's enhancement of reaction kinetics. Together, these factors contribute to the high AEE observed, with TMAF delivering an impressive EE of 95.2%. Furthermore, Fig. 6e highlights a reduction in battery polarization potential, attributed to minimized side reactions (such as self-discharge) and improved reaction kinetics. The initial charge-discharge curves in the inset of Fig. 6e display more symmetrical profiles with TMAX, indicative of enhanced reaction reversibility.

The TMAX electrolytes also improved rate performance, enhancing CE and capacity across specific currents from 0.05 to 1 A g$^{-1}$ (Fig. 6f, Supplementary Fig. 52). At 1 A g$^{-1}$, the TMAX electrolyte enabled stable cycling for over 10,000 cycles, achieving an ACE of 99.8% (Br) and 99.9% (F, Cl). Notably, TMAF achieved a minimal capacity degradation of 0.1‰ per cycle and a high EE of 94.8%, setting a high EE for Zn-halogen batteries (Fig. 6g, h, Supplementary Table 5)[22,54–62]. For commercial relevance, we assembled ZICBs with 10.06 mg of active $I_2$ (based on I$^-$ concentration) and tested them at a low specific current of 0.05 A g$^{-1}$ using TMAF electrolyte. Despite the shuttle effect's greater impact at low specific currents, the ZICBs achieved an impressive ACE of 98.6%, a capacity of 0.47 mAh cm$^{-2}$, and a high capacity retention of 97.22% after 500 cycles (Supplementary Figs. 53, 54).

Exploring higher charging potentials for increased capacity, we found that raising the upper limit to 1.8 V resulted in unstable cycling and only 52.9% EE. When the potential limit was set to 2 V, the charging potential could only reach approximately 1.85 V, which is attributed to irreversible side reactions in the electrolyte at high potentials[63–66]. (Supplementary Fig. 55 and Supplementary Table 6) Fortunately, when the upper potential was maintained at 1.6 V, the EE value remained consistently high (95.5%). While higher potentials may increase capacity, prioritizing high EE over maximum capacity is essential for sustainable battery operation, as excessive capacity gain at the expense of EE leads to greater resource waste and operational costs. This study underscores the importance of balancing high-energy output with high efficiency in energy storage solutions, as both are vital for sustainable and economically viable battery technology.

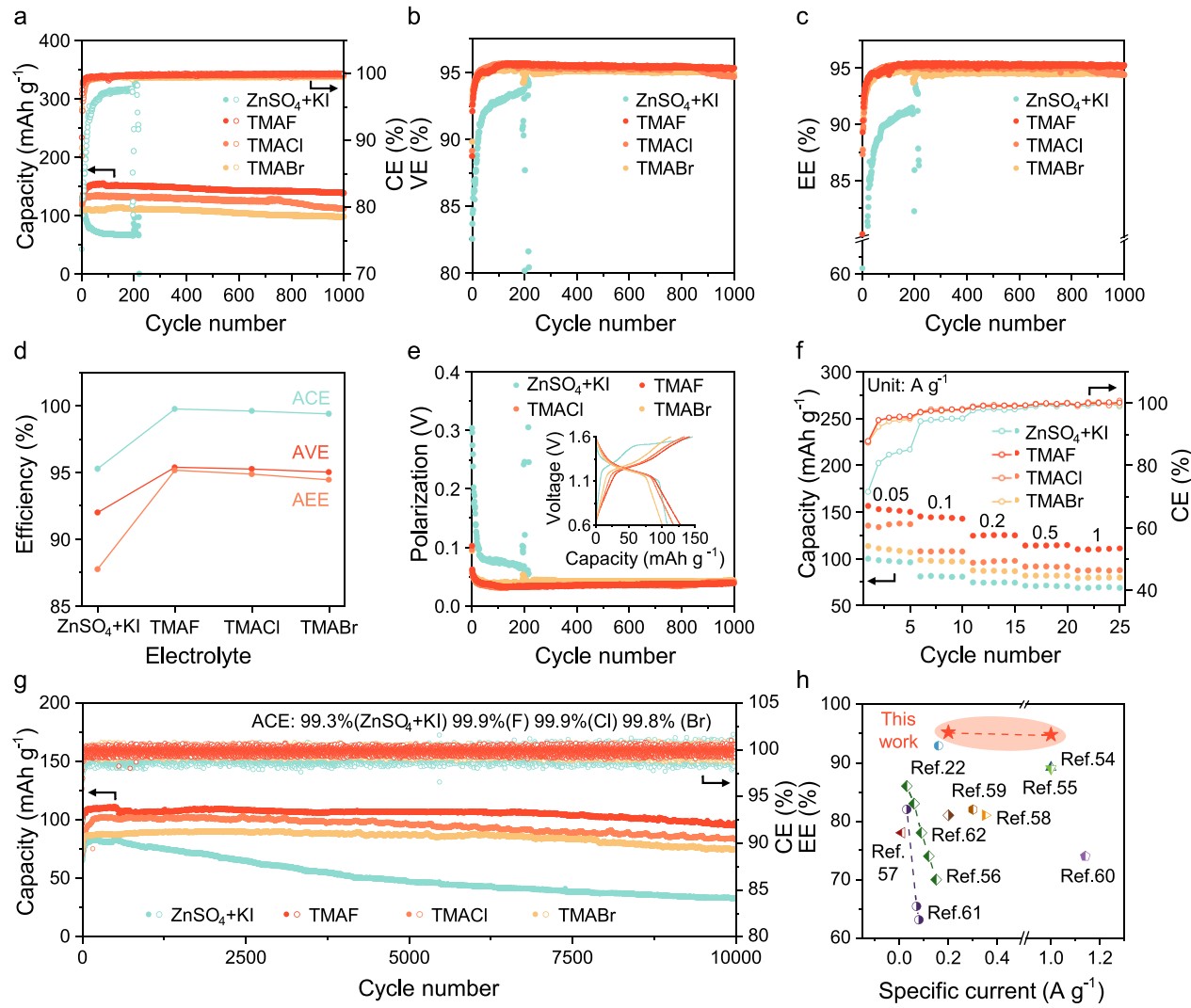

**Fig. 6 | Electrochemical performance of ZICBs. a** Cycling stability and CE, **b** VE, and **c** EE profiles of ZICBs using different electrolytes at a specific current of 0.2 A g⁻¹, tested at 30 °C. **d** ACE, AVE and AEE in different electrolytes. **e** Polarization voltage of ZICBs using different electrolytes at a specific current of 0.2 A g⁻¹, inset is charge/discharge curves of the first cycle. **f** Rate performance. **g** Cyclic stability and CE at a specific current of 1 A g⁻¹, tested at 30 °C. **h** Comparison of the energy efficiency of ZICBs with previous reports. Source data are provided as a Source data file.

## Discussion

In conclusion, this study introduces a cation-driven phase transition to suppress the shuttle effect and an innovative anion-enhancement strategy to enhance reaction kinetics. The TMA⁺ cation in the TMAX additive demonstrates a strong complexing ability to capture soluble $I_3^-$ ions, forming a solid-phase interhalide complex that immobilizes polyiodide ions through a liquid-solid phase transition, effectively suppressing the shuttle effect. Additionally, TMA⁺ preferentially adsorbs onto the Zn negative electrode, creating a cationic electrostatic shielding layer that mitigates dendrite formation and promotes uniform and dense Zn²⁺ deposition, significantly enhancing Zn negative electrode stability. The halogen anion X⁻ (X = F, Cl, Br) in the additives reduces the ΔG of $I^- \rightarrow I_2X^-$ and $I_2X^- \rightarrow TMAI_2X$, facilitating $I^-/I_2X^-/TMAI_2X$ conversions and thereby making the reaction more accessible. The kinetic enhancement effect follows the trend F > Cl > Br. This combined action of the TMA⁺ cation and X⁻ anion addresses key challenges: positive electrode active material dissolution, negative electrode dendrite formation, and self-discharge. These improvements stabilize the interface between the electrodes, increase battery capacity and cycle life, and achieve high energy efficiency. With TMAF-modified electrolytes, the ZICB's CE improved from 80.95% to

98.32% in a 60 h self-discharge experiment. At a low specific current of 0.2 A g⁻¹, ZICBs achieved an AEE of up to 95.2%, demonstrating robust reversibility with a capacity degradation of only 0.1% per cycle over 1000 cycles. At a specific current of 1 A g⁻¹, the ZICBs maintained a low capacity degradation of just 0.1‰ per cycle over 10,000 cycles. This pioneering anion-enhancement strategy effectively addresses the challenges of EE and reversibility, providing a promising pathway for sustainable and high-performance energy storage development.

## Methods

### Materials

ZnSO₄·7H₂O (analytical reagent, AR) was procured from Aladdin. KI (99.0%), Iodine powder (99.8%), (CH₃)₄NF·4H₂O (TMAF·4H₂O, 97%), (CH₃)₄NCl (TMACl, AR) and (CH₃)₄NBr (TMABr, AR) were acquired from Macklin. Activated carbon (YP80F) and carbon paper (HCP030N) was purchased from Guangdong Canrd Technology Co., Ltd.

### Electrolyte Preparation

The electrolyte consists of two parts: the negolyte and the posolyte. The negolyte is prepared by dissolving 2 mol L⁻¹ (M) ZnSO₄ in deionized water. The blank electrolyte and the modified electrolyte,

serving as the posolyte, are prepared by dissolving 0.4 M KI in deionized water, without or with the addition of 0.2 M TMAX (where X = F, Cl, Br), respectively. The electrolyte preparation was carried out at $25 \pm 1$ °C. The combinations of the negolyte with different posolytes are denoted as $ZnSO_4$ + KI and TMAX. In practical applications, the volume ratio of the negolyte to the posolyte is 10:3.

## Preparation of $KI_3$ solution
Add 1.6600 g of KI to 10 mL of deionized water and stir until fully dissolved. Then, introduce 0.1524 g of $I_2$ into the resulting solution and stir for approximately 10 minutes.

## Preparation of positive electrode current collectors
Activated carbon (YP80F) and polyvinylidene fluoride (PVDF5130, $M_w$ = 1100000 Da) are mixed in a mass ratio of 9:1, using N-methyl-2-pyrrolidone (NMP) as the dispersant. The mixed slurry is coated onto carbon paper using a squeegee and dried under vacuum at 60 °C for 10 h to prepare the positive electrode current collector. The mass loading of activated carbon on the current collector is 2-3 mg cm$^{-2}$.

## Electrochemical measurements
CR2032 stainless steel coin-type cells were assembled under ambient conditions. Zn||Zn symmetric cells use Whatman GF/D as the separator, while ZICBs employ Whatman GF/A as the separator. The diameter of the Whatman GF/A separator used is 16 mm, with a thickness of 260 μm and an average pore size of 1.6 μm, while the Whatman GF/D separator also has a diameter of 16 mm, a thickness of 675 μm, and an average pore size of 2.7 μm. The zinc foil used has a lateral diameter of 11 mm and a thickness of 100 μm. The total volume of electrolyte used for GF/A separator is 130 μL and for GF/D separator is 200 μL. ZICBs are assembled in the following order: Zn foil, negolyte, separator, posolyte, and activated carbon electrode. The volume ratio of the negolyte to posolyte used is 10:3. Theoretically, the mass of $I_2$ generated during charging in the positive electrode is 1.524 mg cm$^{-2}$. The electrochemical performance of ZICBs was studied using the LAND Electrochemical Testing System and Bio-logic Electrochemical Workstation in the thermostatic chamber at $30 \pm 1$ °C. The Tafel slope was obtained from linear sweep voltammetry (LSV) in ZICBs with a scan rate of 2 mV s$^{-1}$. Cyclic voltammetry (CV) was conducted at a scan rate of 1–5 mV s$^{-1}$. Electrochemical impedance spectroscopy (EIS) was performed in the frequency range of 100 kHz–0.1 Hz with a voltage amplitude of 10 mV, and the in situ EIS test was performed at 0.2 A g$^{-1}$ after 1 h of rest. During the galvanostatic intermittent titration technique (GITT) test, a specific current pulse of 0.05 A g$^{-1}$ is applied for 10 min, followed by a 30 min rest period. The test potential range is 1.6–0.6–1.6 V.

## Material characterizations
The UV-vis spectra were acquired using a UV-Vis Spectrophotometer (UV-2600i). Raman spectra were carried out on a Horiba LabRAM HR Evolution. Fourier transform infrared spectra (FT-IR) were performed on a Thermo Fisher Scientific Nicolet iS20. The scanning electron microscopy (SEM) images of the positive electrode and Zn were obtained using a Hitachi SU-8010 SEM at an acceleration voltage of 3 kV. X-ray photoelectron spectroscopy (XPS) analyses were carried out on a Thermo Scientific ESCALAB 250Xi. All binding energies were referenced to the C 1s peak (284.8 eV). Time of flight secondary ion mass spectrometry (ToF-SIMS) measurements were conducted on an ION-TOF GmbH TOF SIMS 5 instrument. High resolution transmission electron microscope (HRTEM) and energy dispersive spectrometer (EDS) measurements were conducted on a Talos F200X. Focused Ion Beam-Scanning Electron Microscope (FIB-SEM) measurements were performed on a Helios G4 PFIB. Nitrogen ($N_2$) adsorption experiments were conducted on Micromeritics ASAP 2460. X-ray diffraction (XRD)

patterns and 2-dimensional grazing-incidence wide-angle X-ray scattering (2D-GIWAXS) patterns were recorded using an Xeuss 2.0 with Cu Kα radiation (λ = 1.5406 Å). Electron backscatter diffraction (EBSD) results were acquired using an EDAX Hikari Plus. The 3D morphology and surface roughness were obtained through the confocal laser scanning microscope VK-X3000.

## Density functional theory (DFT) calculation
The geometry of $I_2{\cdots}X^-$ (X = F, Cl, Br and I) were optimized using M06-2X density functional theory[67] method with Def2-TZVPP basis set. Vibrational contribution to Gibbs free energy and zero-point energy were calculated at the same level. All these calculations were performed using the Gaussian 16 program. The adsorption energy calculations were performed using the Vienna Ab Initio Simulation Package (VASP)[68] with the projector augmented wave (PAW) method[69]. The exchange-correlation interactions were modeled with the Perdew-Burke-Ernzerhof (PBE) functional[70] within the generalized gradient approximation (GGA) framework. A plane-wave basis set with an energy cutoff of 500 eV was employed, and self-consistent field iterations were carried out until the total energy converged to a threshold of 10$^{-6}$ eV. The force on each atom less than 0.01 eV/Å was set for the convergence criterion of geometry optimization. The Brillouin zone integration was performed using $2 \times 2 \times 1$ k-point sampling.

## Data availability
All data that support the findings of this study are presented in the manuscript and Supplementary Information. Source data are provided with this paper.

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

## Acknowledgements

This work is supported by the National Natural Science Foundation of China (22408324), the Zhejiang Provincial Natural Science Foundation of China (LQ24B030002), and the China Postdoctoral Science Foundation (2022M722729, 2023T160571).

## Author contributions

Conceptualization: W.Z., H.C. and Y.L. Experimental design and investigation: W.Z. and H.C. Data analyses: W.Z., H.C., S.Z., L.L., C.T., W.C. and Y.L. Calculation: H.C. Writing-original draft: W.Z. Writing-review & editing: W.Z., H.C. and Y.L.

## Competing interests

The authors declare no competing interests.
