## [Peer Review File · Nature Communications]

Cation-Driven Phase Transition and Anion-Enhanced Kinetics for High Energy Efficiency Zinc-Interhalide Complex Batteries

Corresponding Author: Professor Yingying Lu

Version 0:

Reviewer comments:

Reviewer #1

(Remarks to the Author)

This work by Lu et al. reports a cation-driven phase transition to suppress the shuttle effect, along with an innovative anion-enhancement strategy to enhance reaction kinetics, thereby achieving pioneering energy efficiency. It presents interesting conclusions, highlighting the significant role of F chemistry in enhancing reaction pathways and achieving high energy efficiency, which could inspire future researchers. The investigation into the mechanisms of electrode-electrolyte interface is comprehensive and meticulous, with an impressive utilization of electron backscatter diffraction to visually demonstrate the crystallographic orientation of Zn. In general, this work is innovative and crucial, which could provide insights into Zn-halogen batteries and the broader aqueous battery community. I therefore recommend acceptance of this manuscript in Nature Communications after the issues listed as follows are properly addressed by the authors.

1. In Fig. 4a, why do symmetric cells in TMAX electrolyte exhibit a larger overpotential during the initial cycles compared to those in ZnSO₄+KI electrolyte? It is suggested to clarify the reason.
2. The article mentions that "When the voltage limit was set to 2 V, the charging voltage could only reach approximately 1.85 V, which is attributed to irreversible side reactions in the electrolyte at high voltages". In this context, what do the "irreversible side reactions in the electrolyte at high voltages" specifically refer to? It is recommended to provide detailed clarification by referring to additional references.
3. The author introduced fluorine components into the electrolyte. Does this lead to any changes in the SEI layer on the zinc anode? Please provide some characterization and discussion regarding the SEI.
4. Reversibility of Zn anode is highly related to testing conditions (10.1016/j.joule.2024.07.023), it should be discussed further to highlight the testing conditions and what will happen in more critical conditions. I suggest a unsymmetric batteries should be tested until the inflection point of CE appears.
5. It is claimed that "the VE enhancement stems from TMAX's enhancement of reaction kinetics", how can TMAX's can enhance kinetics of the reaction. There are a few steps for ion transportation and reaction; which one is the rate determined step, and which one can be improved by TMAX and why?
6. Apart from improving short-circuit resistance, what geometric structural advantages does the 101-oriented growth have compared to the traditional 002-oriented growth?

Reviewer #2

(Remarks to the Author)

Aqueous zinc-halogen batteries have shown considerable promise as a high-safety energy storage system. Nevertheless, they currently encounter challenges including low energy efficiency and restricted cycle life. The authors proposed an effective strategy to simultaneously address zinc deposition issues at the anode and shuttle effects caused by halogen species at the cathode. In addition, this strategy offers detailed scientific insights into the underlying mechanisms. This work can be accepted in Nature Communications after addressing the following points.

- (1) Does TMAF contribute to the formation of a ZnF₂ layer at the interface?
- (2) A series of electrochemical tests such as Tafel analysis and LSV should be conducted to comprehensively evaluate the

effects of TMAX on zinc deposition as well as its influence on other behaviors and performance.

(3) In symmetric cell testing, it is suggested to provide additional data (such as EIS) to eliminate potential interference from soft short circuits.

(4) In the introduction, when describing the criteria to be met by ligands capable of trapping polyiodide intermediates, it is mentioned that "they contain atoms or groups with lone pair electrons (e.g., N, O, S) to ensure strong bonding with polyiodides". In fact, it is the positive charge of N⁺ rather than the lone pair of electrons that plays a major role in complexing agents such as quaternary ammonium salts. Please check for misrepresentation here.

(5) The authors should clearly define the active material used for calculating capacity in the manuscript.

(6) It appears that an error occurred during the conversion of the article from a Word document to a PDF document, which has led to issues in lines 540-606. It is uncertain whether any information was omitted. Therefore, it is recommended to conduct a thorough review of this section.

Reviewer #3

(Remarks to the Author)

The authors present a cation-driven phase transition approach to mitigate the shuttle effect and an anion-enhancement strategy to improve reaction kinetics in Zn-interhalide complex batteries. Tetramethyl quaternary ammonium cation (TMA⁺) is introduced as an additive that immobilizes polyiodide ions by forming solid-phase interhalide complexes, thereby reducing self-discharge. The authors propose that anionic components (X⁻ = F, Cl, Br) influence reaction kinetics, with F⁻ demonstrating superior performance. Additionally, TMA⁺ adsorbs onto the Zn anode to create an electrostatic shield, promoting the formation of a Zn (101) crystal surface texture.

Although the results are supported by various analytical tools, the manuscript lacks detailed scientific explanations for the observed behaviors and outcomes. Furthermore, the work does not demonstrate sufficient novelty to meet the high standards required for publication in Nature Communications.

1. Quaternary ammonium compounds are well-established as solid-complexing agents and have been widely utilized to suppress the cross-diffusion of halide catholytes, particularly in zinc-bromine batteries. In this context, the manuscript lacks adequate citation of relevant prior work, such as the use of tetrabutylammonium (TBA) as a solid-complexing agent in zinc-bromine batteries and TBA's role as a surface adsorption agent in zinc-ion batteries, both of which have already been reported.

2. "...this liquid-solid conversion often exhibits slow reaction kinetics, impacting reversibility."

According to this draft, TMAI₂F is the most thermodynamically stable, demonstrating the exceptional ability of TMAF to capture polyiodine. However, how do these results indicate that TMAF effectively boosts the kinetic performance of both oxidation and reduction processes, especially in battery applications related to discharge power output? How does the solid complex formation correlate with the electrochemical reversibility?

3. Please provide experimental details on how to chemically synthesize the KI₃ solution exclusively and how to characterize its formation.

4. The detailed information on activated carbon (e.g., surface area, pore volume) is missing. Is there any correlation between the size of the complex and the pores of the activated carbon in retaining charged products? Does TMAX become confined within the pores, or does it simply cover the surface?

5. According to the manuscript, the polyiodide ion capture efficacy follows the order: TMAF > TMACl > TMABr. What is the main reason that among I₂X⁻ species, I₂F⁻ exhibits a more uniform distribution on the activated carbon electrode?

6. "...These comprehensive analyses confirm the formation and uniform distribution of interhalide complexes TMAI₂X on the electrode, highlighting TMAX's role in stabilizing the electrode surface."

What is meant by "stabilizing"? Isn't this simply deposition?

7. Solid bromine complexes are not formed with short-alkyl-chain tetramethyl and tetraethyl ammonium salts. Why, then, is a solid complex formed with TMA⁺ in this case?

8. Zn||Zn symmetric cells use Whatman GF/D as the separator, while Zn-interhalide complex batteries (ZICBs) employ Whatman GF/A. Is there a specific reason for this choice? Does it significantly affect cell performance?

Version 1:

Reviewer comments:

Reviewer #1

(Remarks to the Author)

Reading through the revised manuscript, I think the paper has been well improved and all my concerns have been well addressed. The paper now can be accepted as it is.

Reviewer #2

(Remarks to the Author)

The paper can be accepted without changes.

Reviewer #3

(Remarks to the Author)

The authors have addressed all questions and comments in an impressive manner.

Responses letter

Dear Editors and Reviewers:

Thank you for your letter and for the reviewers' comments concerning our manuscript titled "Cation-Driven Phase Transition and Anion-Enhanced Kinetics for Ultra-High Energy Efficiency Zn-Interhalide Complex Batteries". Those comments are all valuable and very helpful for improving our paper. We believe that our explanations have now clarified the issues raised by reviewers. All changes are included in the revised manuscript with marked colors for the related reviewers. Responses to the reviewers' comments are as following.

Reviewer #1:

This work by Lu et al. reports a cation-driven phase transition to suppress the shuttle effect, along with an innovative anion-enhancement strategy to enhance reaction kinetics, thereby achieving pioneering energy efficiency. It presents interesting conclusions, highlighting the significant role of F chemistry in enhancing reaction pathways and achieving high energy efficiency, which could inspire future researchers. The investigation into the mechanisms of electrode-electrolyte interface is comprehensive and meticulous, with an impressive utilization of electron backscatter diffraction to visually demonstrate the crystallographic orientation of Zn. In general, this work is innovative and crucial, which could provide insights into Zn-halogen batteries and the broader aqueous battery community. I therefore recommend acceptance of this manuscript in Nature Communications after the issues listed as follows are properly addressed by the authors.

Q1. In Fig. 4a, why do symmetric cells in TMAX electrolyte exhibit a larger overpotential during the initial cycles compared to those in ZnSO₄+KI electrolyte? It is suggested to clarify the reason.

Response: Thanks for the reviewer's constructive question. The experimental

result that the symmetric cell in TMAX electrolyte exhibits a higher overpotential during the initial cycling compared to the symmetric cell in ZnSO₄+KI electrolyte is closely related to the adsorption of TMA⁺ ions on the zinc electrode surface (Adv. Energy Mater. 2022, 12, 2102780). As shown in Fig. 4g of the manuscript, DFT calculations have demonstrated that TMA⁺ preferentially adsorbs on the zinc anode surface. During the initial cycling of the battery, the deposition of zinc does not initiate at specific sites, but rather occurs more uniformly across the electrode surface due to the electrostatic shielding effect of TMA⁺ ions. This process requires a higher voltage to overcome the electrostatic potential barrier on the electrode surface, leading to an elevated initial overpotential (Adv. Energy Mater. 2023, 13, 2203254). In conjunction with the nucleation overpotential results in Supplementary Fig. S43, the symmetric cell in TMAX electrolyte exhibits a higher nucleation overpotential. This higher energy barrier implies that a stronger driving force is required during the initial nucleation stage, which promotes more uniform zinc deposition and results in finer grain sizes. This is beneficial for suppressing dendrite formation (Adv. Energy Mater. 2022, 12, 2103231; J. Am. Chem. Soc. 2023, 145, 29, 15776-15787).

Q2. The article mentions that “When the voltage limit was set to 2 V, the charging voltage could only reach approximately 1.85 V, which is attributed to irreversible side reactions in the electrolyte at high voltages”. In this context, what do the “irreversible side reactions in the electrolyte at high voltages” specifically refer to? It is recommended to provide detailed clarification by referring to additional references.

Response: Thanks for the reviewer’s constructive suggestion. The “irreversible side reactions in the electrolyte at high voltages” mentioned in the text are primarily related to IO₃⁻. There are several studies available that can support this claim. Studies have shown that in aqueous electrolytes, the formation and consumption of IO₃⁻ typically occur when the voltage is around 2 V. During charging, when the voltage reaches approximately 2.21 V (vs. Zn²⁺/Zn),

reactions " $\frac{1}{2}\text{I}_2+\text{H}_2\text{O}-\text{e}^-\leftrightarrow\text{HIO}+\text{H}^+$ " and " $5\text{HIO}\rightarrow\text{IO}_3^-+2\text{I}_2+2\text{H}_2\text{O}+\text{H}^+$ " take place, leading to the formation of IO_3^- . During discharge, when the voltage is around 1.84 V (vs. Zn^{2+}/Zn), reaction " $\text{IO}_3^-+6\text{H}^++6\text{e}^-\leftrightarrow\text{I}^-+3\text{H}_2\text{O}$ " is theoretically expected to occur. However, in practice, after the formation of IO_3^- , the presence of I^- in the electrolyte typically triggers reaction " $5\text{I}^-+\text{IO}_3^-+6\text{H}^+\rightarrow 3\text{I}_2+3\text{H}_2\text{O}$ ", causing the conversion of IO_3^- to become irreversible (Nat. Energy 2024, 9, 714-724; Adv. Energy Mater. 2023, 13, 2301049; ACS Nano 2024, 18, 28557-28574; Adv. Mater. 2024, 36, 2405473). To enhance the credibility of the conclusions, these references have been included in the results and discussion section.

Results and discussion

When the voltage limit was set to 2 V, the charging voltage could only reach approximately 1.85 V, which is attributed to irreversible side reactions in the electrolyte at high voltages.⁶³⁻⁶⁶

65. Xie, C. et al. Reversible multielectron transfer I^-/IO_3^- cathode enabled by a hetero-halogen electrolyte for high-energy-density aqueous batteries. *Nat. Energy* **9**, 714-724 (2024).

66. Kushwaha, R. et al. Made to Measure Squaramide COF Cathode for Zinc Dual-Ion Battery with Enriched Storage via Redox Electrolyte. *Adv. Energy Mater.* **13**, 2301049 (2023).

Q3. The author introduced fluorine components into the electrolyte. Does this lead to any changes in the SEI layer on the zinc anode? Please provide some characterization and discussion regarding the SEI.

Response: We thank the reviewer for this valuable suggestion. To investigate the composition of the solid electrolyte interphase (SEI) layer on the zinc anode, we assembled a Zn||Cu asymmetric battery and cycled it for 10 cycles. After

cycling, the Cu electrode, which had zinc deposition, was extracted and subjected to X-ray photoelectron spectroscopy (XPS) analyses. The corresponding XPS spectra results for the electrodes in ZnSO₄+KI and TMAX electrolytes are shown in Figs. R1-R4, respectively.

For both electrodes prepared in the ZnSO₄+KI and TMAX electrolytes, the Zn 2p XPS spectra revealed two peaks at 1021.8 eV for the Zn 2p_{3/2} orbital, and at 1044.8 eV for the Zn 2p_{1/2} orbital, corresponding to the Zn²⁺ (Figs. R1-R4a, *Energy Environ. Sci.*, 2025, DOI:10.1039/D4EE06048B). In the S 2p XPS spectra of Figs. R1-R4b, two peaks at 168.1 eV and 169.0 eV are observed, which are associated with SO₄²⁻. Additionally, the peak at 531.6 eV in the O 1s XPS spectra of Figs. R1-R4c is related to the S=O bond, further confirming the presence of SO₄²⁻ (*Energy Environ. Sci.*, 2025, DOI: 10.1039/D4EE03750B). Since no Zn-OH signals were observed on the Zn 2p XPS spectra (theoretically near 1025.0 and 1048.1 eV) and no OH signals were observed on the O 1s XPS spectra, it can be ruled out that the signals of Zn²⁺ and SO₄²⁻ are derived from Zn₄SO₄(OH)₆·5H₂O (*Energy Environ. Sci.*, 2024,17, 2059-2068; *Energy Environ. Sci.*, 2025, DOI: 10.1039/D4EE03750B). The Zn²⁺ and SO₄²⁻ signals could originate from the residual ZnSO₄ electrolyte on the surface. Unusually, in the N 1s XPS spectra of Figs. R2-R4d, weak N-C signals appeared on the electrodes with the TMAX electrolyte, which was related to the adsorption of a small amount of TMA⁺ on the Zn surface (*Angew. Chem. Int. Ed.* 2024, 63, e202409957). In the F 1s, Cl 2p and Br 3d XPS spectra in Figs. R2-R4e, no corresponding signal peaks appeared, suggesting that the halogen ions in TMAX are not involved in the composition of SEI on the Zn surface. Therefore, it can be concluded that the introduction of the F component does not lead to the alteration of the SEI layer on the Zn anode surface.

Fig. R1. XPS spectra of Cu electrode with zinc deposited in ZnSO₄+KI electrolyte.

Fig. R2. XPS spectra of Cu electrode with zinc deposited in TMAF electrolyte.

Fig. R3. XPS spectra of Cu electrode with zinc deposited in TMACl electrolyte.

Fig. R4. XPS spectra of Cu electrode with zinc deposited in TMABr electrolyte.

Q4. Reversibility of Zn anode is highly related to testing conditions (10.1016/j.joule.2024.07.023), it should be discussed further to highlight the testing conditions and what will happen in more critical conditions. I suggest a unsymmetric batteries should be tested until the inflection point of CE appears.

Response: We thank the reviewer for this valuable suggestion. This work is outstanding and meticulous. It is of great significance for us to evaluate zinc anode reversibility.

To accurately and directly determine the reversibility of the Zn anode, we assembled Zn||Cu asymmetric cells using ZnSO₄+KI and TMAF electrolytes, respectively, and tested them under high areal capacity conditions of 10 mA cm⁻² and 10 mAh cm⁻² (zinc utilization rate, ZUR = 17%). The results are shown in Supplementary Figs. S35-36. In the ZnSO₄+KI electrolyte, the Zn||Cu asymmetric cells exhibited a limited cycle life of merely 33 cycles with an average Coulombic efficiency (CE) of 99.73%, followed by rapid failure due to short-circuiting. This premature failure mechanism, observed prior to zinc depletion, primarily stems from the high areal capacity triggering excessive zinc consumption. Such accelerated zinc utilization promoted rapid byproduct accumulation and pronounced dendrite formation, thereby inducing separator penetration and short-circuiting without exhibiting the characteristic capacity inflection point, which underscores the poor reversibility of the Zn anode under high-capacity operating conditions (Supplementary Fig. S35).

Supplementary Fig. S35. **a** Capacity-CE curves of a Zn||Cu asymmetric cell using ZnSO₄+KI electrolytes at 10 mA cm⁻² and 10 mAh cm⁻². **b** Corresponding

time-voltage curve. Voltage profiles of the **c** 1st, **d** 30th, and **e** 34th cycles.

In the TMAF electrolyte, the Zn||Cu asymmetric cells maintained stable cycling for 212 cycles with an average CE of 99.78%—more than 6 times longer than with ZnSO₄+KI (Supplementary Fig. S36). In Supplementary Fig. S36b, a progressive intensification of voltage polarization at the end of discharge was observed at the 213th cycle, corresponding to the inflection point in the capacity curve depicted in Supplementary Fig. S36a. This phenomenon is attributed to the depletion of active Zn, which induced a drastic rise in internal resistance and voltage polarization. The 213th cycle delivered an areal capacity of 9.947 mAh cm⁻², equivalent to a Zn thickness of 16.91 μm. When utilizing a 100 μm Zn anode, 212 cycles resulted in a cumulative consumption of 83.91 μm Zn, corresponding to an average consumption of 0.396 μm Zn per cycle (0.232 mAh cm⁻²). These data reveal that the actual efficiency of Zn participation in the primary reaction reached $(10-0.232)/10 \times 100\% = 97.68\%$ per cycle on average. Such high efficiency unambiguously demonstrates the exceptional reversibility of the Zn anode in the TMAF electrolyte system, as the majority of Zn utilization was governed by the reversible plating/stripping processes rather than irreversible parasitic reactions. This conclusion has been supplemented in the Results and discussion section of the manuscript.

Supplementary Fig. S36. **a** Capacity-CE curves of a Zn||Cu asymmetric cell using TMAF electrolytes at 10 mA cm^{-2} and 10 mAh cm^{-2} . **b** Corresponding time-voltage curve. Voltage profiles of the **c** 1st, **d** 210th, and **e** 213th cycles.

Q5. It is claimed that “the VE enhancement stems from TMAX’s enhancement of reaction kinetics”, how can TMAX’s can enhance kinetics of the reaction. There are a few steps for ion transportation and reaction; which one is the rate determined step, and which one can be improved by TMAX and why?

Response: We thank the reviewer for this valuable and constructive question. Ion transportation and reaction primarily involve two steps (Fig. 3k): **Step I:** $2\text{I}^- - 2\text{e}^- + \text{X}^- \rightarrow \text{I}_2\text{X}^-$, I^- and X^- combine undergo electron loss to form I_2X^- ; **Step II:** $\text{TMA}^+ + \text{I}_2\text{X}^- \rightarrow \text{TMAI}_2\text{X}$, TMA⁺ combines with I_2X^- to form solid TMAI₂X.

To investigate the influence of TMAX on the reaction kinetics, density functional theory (DFT) was employed to calculate the Gibbs free energy differences (ΔG) for step I and step II, with the results presented in Fig. 3g and Supplementary Table S3. The ΔG values for both step I and step II follow the trend $\text{F} < \text{Cl} < \text{Br} < \text{I}$, with the ΔG for step I being lower than that for step II. A lower ΔG typically corresponds to a stronger thermodynamic driving force, which favors an

increased reaction rate and better kinetic performance (Nat. Commun. 2023, 14, 1856; Energy Environ. Sci., 2023,16, 4630-4640; Energy Environ. Sci., 2023, 16, 4073–4083; Adv. Funct. Mater. 2024, 2422868; Nano Energy 2025, 133, 110519; Adv. Sci. 2024, 11, 2410653). Since ΔG in step II is higher, it indicates that this step is more likely to correspond to a slower reaction rate. Additionally, step II involves a liquid-solid phase transition, and phase transition processes typically require overcoming high activation energy barriers, which leads to a significant reduction in the reaction rate (Nat. Mater. 2012, 11, 952-957; Laidler, K. J., & Meiser, J. H. (1982). Physical Chemistry (3rd ed.). Benjamin/Cummings). This results in a higher kinetic resistance, making it the likely rate determined step. Based on these two reasons, **step II can be considered the rate determined step.**

The improvement in kinetic performance by TMAX is achieved through a two-step mechanism (Fig. 3k). In step I, I^- and X^- combine and undergo electron loss to form I_2X^- , where X^- reduces the reaction's ΔG (with F having the most pronounced effect), making the redox process thermodynamically favorable and accelerating the reaction rate. In step II, TMA^+ combines with I_2X^- to form solid $TMAI_2X$, where the ΔG of F is the lowest, providing a stronger driving force that enhances the kinetic performance. Therefore, **TMAF plays a crucial role in enhancing the reaction kinetics in both steps**, contributing to the improvement of VE.

Detailed information is provided in the Results and discussion section of the manuscript.

Halide Anion-Enhanced Cathode Reaction Kinetics

To explain the kinetic differences among halide ions, density functional theory (DFT) calculations were performed. Gibbs free energy differences (ΔG) for reactions of $I^- \rightarrow I_2X^- \rightarrow TMAI_2X$ ($X = F, Cl, Br, I$) were calculated (Fig. 3g and Supplementary Table S3). The ΔG values for $I^- \rightarrow I_2X^-$ follow the trend $I_2F^- < I_2Cl^- < I_2Br^- < I_3^-$. Lower ΔG values indicate higher reaction spontaneity, with I_2F^- being

the most favorable and stable due to its lowest ΔG . This suggests that TMAF facilitates the acceleration of the reaction rate and enhances the iodide redox kinetics.⁴⁶⁻⁴⁸ Structural analysis (Fig. 3h) revealed that all I_2X^- species exhibit linear or near-linear trihalide structures.⁴⁹ Among them, I_2F^- has the shortest I-X bond length, indicating the strongest binding affinity, which decreases in the order $F > Cl > Br > I$. Electrostatic potential (ESP) distributions (Fig. 3i and Supplementary Fig. S27) show that iodine atoms in I_3^- carry negative charges, while bonding with electronegative halogens (F, Cl, Br) causes electron transfer, resulting in the I atom adjacent to X^- carrying a partial positive charge. F, with its high electronegativity, induces the highest positive charge on the adjacent I atom. These findings are consistent with XPS observations for the cathode electrode (Figs. 2g-i and Supplementary Figs. S14, S15).

The ΔG values for the reaction $I_2X^- \rightarrow TMAI_2X$ also follow the trend: $TMAI_2F < TMAI_2Cl < TMAI_2Br < TMAI_3$ (Fig. 3g). Lower ΔG values favor the reaction, indicating that $TMAI_2F$ is more easily formed and exhibits superior kinetic performance. We also calculated the adsorption energy of $TMAI_2X$ on the activated carbon electrode surface (Fig. 3j and Supplementary Figs. S28-31). The adsorption energy of $TMAI_2X$ on the activated carbon electrode follows the trend: $TMAI_2F < TMAI_2Cl < TMAI_2Br < TMAI_3$. Among them, the adsorption energy of $TMAI_2F$ was the lowest, which indicated that it had the strongest adsorption force and was most likely to be attached to the surface of the activated carbon electrode. This stronger adsorption energy implies that $TMAI_2F$ will firmly adhere to the electrode surface and diffuse more rapidly, leading to a more uniform distribution. In contrast, the adsorption energies of the $TMAI_2Cl$ and $TMAI_2Br$ complexes are weaker, which may result in larger adsorption-desorption fluctuations, leading to unstable adsorption on the electrode surface and a more uneven distribution. This is consistent with experimental observations (Figs. 2c, f).

The computational results align with the aforementioned experimental findings, demonstrating that TMAF significantly enhances the reaction kinetics and ensures uniform distribution on the activated carbon electrode surface. The enhancement of the $I^- \rightarrow TMAI_2X$ reaction occurs through a two-step mechanism (Fig. 3k). In Step I (Eq. 3), I^- and X^- combine and undergo electron loss to form I_2X^- , where X^- lowers the reaction's ΔG (with F having the most pronounced effect), making the redox process energetically favorable. In Step II (Eq. 4), TMA^+ combines with I_2X^- to form solid $TMAI_2X$. The $I_2F^- \rightarrow TMAI_2F$ reaction has the lowest ΔG , making it the most easily formed. Therefore, F chemistry plays a dual role in enhancing the entire reaction pathway, contributing to improved reaction kinetics.

Fig. 3 | Kinetic analysis of ZICBs using TMAX electrolyte. a CV profiles of ZICBs using TMAF electrolyte. **b** Relationship between peak current and square root of scan rate. Tafel plots of **c** IOR and **d** IRR. **e** I^- diffusion coefficient

in different electrolytes. **f** DRT curves charging to different voltage. **g** The Gibbs free energy ladder diagram. **h** Configurational structure and bond length of I_2X^- . **i** Valence states (atomic Mullikan charge) and ESP of I_2F^- . **j** Adsorption energy $TMAI_2X$ on carbon layer. **k** Schematic illustration of the halide anion-enhanced behavior.

Supplementary Table S3. Gibbs free energies of I_2X^- and $TMAI_2X$ and Gibbs free energy differences (ΔG) of step I and step II. (Unit: eV)

X	I_2X^-	$TMAI_2X$	Step I (ΔG)	Step II (ΔG)
F	-4.11	-6.57	-4.11	-2.46
Cl	-3.37	-4.74	-3.37	-1.37
Br	-3.12	-4.27	-3.12	-1.15
I	-2.89	-3.84	-2.89	-0.95

Q6. Apart from improving short-circuit resistance, what geometric structural advantages does the 101-oriented growth have compared to the traditional 002-oriented growth?

Response: We thank the reviewer for this valuable question. Compared to the traditional 002-oriented growth, the geometric structural advantages of the 101-oriented growth are primarily manifested in the following two aspects (Adv. Mater. 2024, 36, 2305988; Angew. Chem. Int. Ed. 2024, e202414757; Adv. Energy Mater. 2024, 14, 2304003):

1. Reduced Lattice Distortion: Due to the relatively weak bonding interactions between (002)-Zn and deposited atoms, zinc deposited on a (002) textured substrate tends to deviate from the original lattice orientation, leading to the accumulation of lattice distortion. Additionally, the interlayer interlocking structure of Zn plates causes an uneven electric field distribution at the interface, which promotes dendrite growth at the gaps. As a result, the (002)-Zn deposition cannot sustain continuous epitaxial growth, limiting its capacity to

manage deposited Zn effectively ($< 4 \text{ mAh cm}^{-2}$). In contrast, (101)-Zn benefits from its unique directional guidance and stronger bonding interactions, which facilitate the formation of a dense, vertical epitaxial structure on the anode surface. This structure minimizes lattice distortion during cycling, ensuring stable cycling even at high areal capacities.

2. Lower Grain Boundary Area: (101)-Zn exhibits a significantly lower grain boundary area compared to (002)-Zn. The reduction in grain boundary area, combined with its higher hydrogen evolution reaction (HER) activation barrier (Zn(101) is considered a highly reactive facet, where the rapid ionic reaction kinetics dominate in competition with the H^+/H_2 reactions, unlike the (002) plane), effectively maintains the pH stability at the electrode interface and suppresses side reactions.

In summary, (101)-Zn demonstrates outstanding performance in high-capacity Zn||Zn symmetric batteries by reducing lattice distortion, minimizing grain boundary area, and enhancing the stability of epitaxial growth. These advantages contribute to a significant improvement in cycling life and stability, while simultaneously suppressing side reactions.

Reviewer #2:

Aqueous zinc-halogen batteries have shown considerable promise as a high-safety energy storage system. Nevertheless, they currently encounter challenges including low energy efficiency and restricted cycle life. The authors proposed an effective strategy to simultaneously address zinc deposition issues at the anode and shuttle effects caused by halogen species at the cathode. In addition, this strategy offers detailed scientific insights into the underlying mechanisms. This work can be accepted in Nature Communications after addressing the following points.

Q1. Does TMAF contribute to the formation of a ZnF_2 layer at the interface?

Response: We thank the reviewer for this valuable question. To investigate the composition of the SEI layer on the Zn anode, we assembled a Zn||Cu asymmetric battery and cycled it for 10 cycles. After cycling, the Cu electrode, which had Zn deposition, was extracted and subjected to XPS and high resolution transmission electron microscope (HRTEM) analyses.

The corresponding XPS spectra results for the electrode in TMAF electrolyte are shown in Fig. R5. For the electrode prepared in TMAF electrolyte, the Zn 2p XPS spectra revealed two peaks at 1021.8 eV for the Zn 2p_{3/2} orbital, and at 1044.8 eV for the Zn 2p_{1/2} orbital, corresponding to the Zn^{2+} (Fig. R5a, Energy Environ. Sci., 2025, DOI:10.1039/D4EE06048B). In the S 2p XPS spectra of Fig. R5c, two peaks at 168.1 eV and 169.0 eV are observed, which are associated with SO_4^{2-} . Additionally, the peak at 531.6 eV in the O 1s XPS spectra of Fig. R5b is related to the S=O bond, further confirming the presence of SO_4^{2-} (Energy Environ. Sci., 2025, DOI: 10.1039/D4EE03750B). Since no Zn-OH signals were observed on the Zn 2p XPS spectra (theoretically near 1025.0 and 1048.1 eV) and no OH signals were observed on the O 1s XPS spectra, it can be ruled out that the signals of Zn^{2+} and SO_4^{2-} are derived from $\text{Zn}_4\text{SO}_4(\text{OH})_6 \cdot 5\text{H}_2\text{O}$ (Energy Environ. Sci., 2024, 17, 2059-2068; Energy Environ. Sci., 2025, DOI: 10.1039/D4EE03750B). The Zn^{2+} and SO_4^{2-} signals

could originate from the residual ZnSO₄ electrolyte on the surface. In the F 1s spectra (Fig. R5d), no peak corresponding to ZnF₂ was observed (theoretically expected at 684.7 eV), indicating that no ZnF₂ layer was formed at the interface. (Energy Environ. Sci., 2025, DOI:10.1039/D4EE06048B; Angew. Chem. Int. Ed. 2025, e202423302; Energy Environ. Sci., 2025, DOI: 10.1039/D4EE03750B)

Fig. R5. **a** Fitted Zn 2p, **b** O 1s, **c** S 2p and **d** F 1s XPS spectra of Cu electrode with Zn deposited in TMAF electrolyte.

In order to observe the composition of the electrode interface, HRTEM was implemented to characterize the electrode surface. The HRTEM images and fast Fourier transform (FFT) insets in Fig. R6 show that the lattice spacing of selected regions I-II is 0.254 nm, corresponding to the (220) lattice plane of

ZnSO₄ (PDF#32-1477), while the lattice spacing of selected region III is 0.209 nm, which corresponds to the (101) lattice plane of Zn (PDF#04-0831). The presence of ZnSO₄ is attributed to the residual ZnSO₄ from the electrolyte. No crystalline planes corresponding to ZnF₂ were observed in the HRTEM images, confirming that no ZnF₂ layer was formed at the interface of the deposited Zn. This conclusion is consistent with the results obtained from XPS characterization, and together, both findings demonstrate that the addition of TMAF did not promote the formation of a surface ZnF₂ layer.

Fig. R6. The HRTEM images of Cu electrode with Zn deposited in TMAF electrolyte. Downside are magnified images of the selected HRTEM regions I-III. Insets are the FFT patterns from the selected HRTEM regions, scale bar=5 1/nm.

Q2. A series of electrochemical tests such as Tafel analysis and LSV should be conducted to comprehensively evaluate the effects of TMAX on zinc deposition as well as its influence on other behaviors and performance.

Response: We thank the reviewer for this valuable suggestion. To investigate

the effect of TMAX on zinc deposition, chronoamperometry (CA) tests were conducted at a constant potential of -150 mV on Zn||Zn symmetric cells using ZnSO₄+KI and TMAF electrolytes (Supplementary Fig. S44). The results showed that in the TMAF electrolyte, Zn exhibited a consistent 3D diffusion process, maintaining a dense and flat surface morphology during the plating process. In contrast, in the ZnSO₄+KI electrolyte, Zn demonstrated a typical 2D diffusion mode with dendritic growth, which is consistent with the SEM results shown in Fig. 4b.

Supplementary Fig. S44. Chronoamperometric curves of Zn electrodeposition in different electrolytes under -150 mV.

To investigate the effect of TMAX on Zn corrosion, Tafel curve and linear sweep voltammetry (LSV) tests were conducted. The Tafel curves (Supplementary Fig. S49) show that the presence of TMAF reduced both the corrosion potential and corrosion current compared to the ZnSO₄+KI electrolyte, indicating that TMAF inhibited hydrogen evolution reaction (HER). Additionally, to better evaluate the inhibitory effect of TMAF on parasitic reactions, LSV tests were performed using a three-electrode configuration. A 2 M Na₂SO₄ solution was used to replace 2 M ZnSO₄ in order to eliminate the interference of Zn²⁺ reduction on the HER measurements (Angew. Chem. Int. Ed. 2023, 62, e202218452). The LSV

results (Supplementary Fig. S50) show that the response current density of the $\text{Na}_2\text{SO}_4+\text{KI}+\text{TMAF}$ electrolyte was significantly lower than that of the $\text{Na}_2\text{SO}_4+\text{KI}$ electrolyte, indicating that TMAF effectively inhibited parasitic reactions. The inhibition of Zn corrosion by TMAF can be attributed to the specific adsorption of TMA^+ cations on the Zn surface, which induces uniform deposition and reduces the number of active sites on the surface. Detailed information is provided in the Results and discussion section of the manuscript.

Supplementary Fig. S49. Tafel plots of Zn electrodes tested in electrolytes (scan rate=1 mV s⁻¹).

Supplementary Fig. S50. LSV characterization in the electrolytes (scan rate=5 mV s⁻¹).

Q3. In symmetric cell testing, it is suggested to provide additional data (such as EIS) to eliminate potential interference from soft short circuits.

Response: We appreciate the valuable suggestions provided by the reviewer. To eliminate the potential interference of soft short circuits on the cycling stability of the symmetric cell, we have magnified the voltage curves of the symmetric cell at different cycles from Fig. 4a and plotted them in Fig. R7. The voltage curves of the symmetric cell in Figs. R7a, b consistently exhibit a sloped profile, indicating normal cycling behavior. To further ensure the absence of soft short circuits, we supplemented electrochemical impedance spectroscopy (EIS) measurements at both the early and late stages of cycling with the TMAF electrolyte in the Zn||Zn symmetric cell. EIS tests were conducted after 5 and 2000 cycles, respectively. The R_{ct} values shown in Figs. R7c, d exhibit a gradual decrease without any sharp drop, indicating stable cycling and confirming the absence of short circuits (Joule 2022, 6, 269–279).

Fig. R7. Enlarged views of galvanostatic cycling stability of symmetric Zn||Zn cells using TMAF electrolytes under 4 mA cm^{-2} , 1 mAh cm^{-2} during **a** 2-8 cycles and **b** 1994-2000 cycles. The impedance spectra of different stages in **c** the early stage (after 5th cycle) of the test and **d** the late stage (after 2000th cycle) of the test.

Q4. In the introduction, when describing the criteria to be met by ligands capable of trapping polyiodide intermediates, it is mentioned that “they contain atoms or groups with lone pair electrons (e.g., N, O, S) to ensure strong bonding with polyiodides”. In fact, it is the positive charge of N⁺ rather than the lone pair of electrons that plays a major role in complexing agents such as quaternary ammonium salts. Please check for misrepresentation here.

Response: Thanks for the reviewer's constructive suggestion. We extend our apologies for the incorrect usage of our initial description. The ligands that have been reported to capture polyiodide intermediates typically contain cationic groups, such as quaternary ammonium salts with N⁺ centers and sulfonium ions with S⁺ centers. (SusMat 2023, 3, 522-532; Adv. Energy Mater. 2024, 2402306; Adv. Sci. 2024, 11, 2305061.) These ligands utilize the positive charge on the cation to attract the negatively charged polyiodide anions through electrostatic interactions, thereby effectively stabilizing the polyiodide species. The reviewer is absolutely correct in pointing out that the positive charge of quaternary ammonium cations (N⁺) plays a dominant role in coordinating polyiodides through electrostatic interactions, rather than lone pair electrons as originally stated. We extend our apologies for the incorrect usage of our initial description and have revised the text accordingly. Specifically, the relevant sentence in the Introduction has been amended to: “Recently, ligands capable of capturing polyiodide intermediates have emerged as promising alternatives, provided they meet the following criteria: (1) they contain cationic groups (e.g., quaternary ammonium ions with N⁺ centers or sulfonium ions with S⁺ centers) to ensure effective stabilization of polyiodides through electrostatic interactions;

(2) they demonstrate excellent solubility and dispersibility in the electrolyte; and (3) they are stable and resistant to decomposition.” We thank the reviewer for this valuable clarification, which significantly improves the precision of our discussion on polyiodide stabilization mechanisms.

Q5. The authors should clearly define the active material used for calculating capacity in the manuscript.

Response: We thank the reviewer for this valuable comment. In this study, all full cells are uniformly calculated using I₂ as the cathode active material. The following equation represents the calculation of the average areal loading of the cathode active material, where n, C, V, m, and M denote the amount of substance, molar concentration, volume, mass, and molar mass, respectively. For the full cell, each coin cell has a cathode area of 1 cm², with 30 μL of catholyte. The catholyte contains 0.4 mol L⁻¹ (M) KI, which is assumed to be fully converted to 0.2 M I₂ during charging.

$$n_{I_2} = C_{I_2} \times V_{\text{catholyte}} = 0.2 \text{ mol L}^{-1} \times 30 \times 10^{-6} \text{ L} = 6.0 \times 10^{-6} \text{ mol}$$

$$m_{I_2} = n_{I_2} \times M_{I_2} = 6.0 \times 10^{-6} \text{ mol} \times 254 \text{ g mol}^{-1} = 1.524 \times 10^{-3} \text{ g} = 1.524 \text{ mg}$$

Therefore, the average areal loading calculated based on I₂ as the active material is 1.524 mg cm⁻². Detailed information is provided in the Methods section.

Q6. It appears that an error occurred during the conversion of the article from a Word document to a PDF document, which has led to issues in lines 540-606. It is uncertain whether any information was omitted. Therefore, it is recommended to conduct a thorough review of this section.

Response: We would like to express our gratitude to the reviewer for the detailed feedback. We have thoroughly reviewed the manuscript and confirmed that there were indeed display issues during the conversion from Word to PDF. The correct original text for lines 540-606 is provided below. We have made the

necessary corrections to the affected sections to ensure that no information was omitted or misrepresented. We appreciate the reviewer for highlighting this issue, and we have ensured the accuracy of the document's content.

Materials

ZnSO₄·7H₂O (analytical reagent, AR) was procured from Aladdin. KI (≥99.0%), I₂ (99.8%), (CH₃)₄NF·4H₂O (TMAF·4H₂O, 97%), (CH₃)₄NCl (TMACl, AR) and (CH₃)₄NBr (TMABr, AR) were acquired from Macklin. Activated carbon (YP80F) and carbon paper (HCP030N) was purchased from Guangdong Canrd Technology Co., Ltd.

Electrolyte Preparation

The electrolyte consists of two parts: the anolyte and the catholyte. The anolyte is prepared by dissolving 2 mol L⁻¹ (M) ZnSO₄ in deionized water. The blank electrolyte and the modified electrolyte, serving as the catholyte, are prepared by dissolving 0.4 M KI in deionized water, without or with the addition of 0.2 M TMAX (where X = F, Cl, Br), respectively. The combinations of the anolyte with different catholytes are denoted as ZnSO₄+KI and TMAX. In practical applications, the volume ratio of the anolyte to the catholyte is 10:3.

Reviewer #3:

The authors present a cation-driven phase transition approach to mitigate the shuttle effect and an anion-enhancement strategy to improve reaction kinetics in Zn-interhalide complex batteries. Tetramethyl quaternary ammonium cation (TMA⁺) is introduced as an additive that immobilizes polyiodide ions by forming solid-phase interhalide complexes, thereby reducing self-discharge. The authors propose that anionic components ($X^- = F, Cl, Br$) influence reaction kinetics, with F^- demonstrating superior performance. Additionally, TMA⁺ adsorbs onto the Zn anode to create an electrostatic shield, promoting the formation of a Zn (101) crystal surface texture.

Although the results are supported by various analytical tools, the manuscript lacks detailed scientific explanations for the observed behaviors and outcomes. Furthermore, the work does not demonstrate sufficient novelty to meet the high standards required for publication in Nature Communications.

Q1. Quaternary ammonium compounds are well-established as solid-complexing agents and have been widely utilized to suppress the cross-diffusion of halide catholytes, particularly in zinc-bromine batteries. In this context, the manuscript lacks adequate citation of relevant prior work, such as the use of tetrabutylammonium (TBA) as a solid-complexing agent in zinc-bromine batteries and TBA's role as a surface adsorption agent in zinc-ion batteries, both of which have already been reported.

Response: We sincerely appreciate the reviewer's insightful feedback regarding the citation of prior work on quaternary ammonium compounds (QACs) in zinc-bromine and zinc-ion batteries.

As correctly pointed out by the reviewer, the application of QACs as bromine complexing agents has indeed been extensively studied in zinc–bromine batteries. QACs such as tetrabutylammonium bromide (TBABr) and tetrapropylammonium bromide (TPABr) have been utilized in zinc–bromine batteries as solid complexing agents to mitigate the cross-diffusion of bromine

(TBABr: *Energy Storage Mater.* 2024, 68, 103331; TPABr: *iScience* 2020, 23, 101348; *Mater. Today* 2024, 80, 353-364). In the introduction, we have discussed some relevant works related to the role of QACs in halogen sequestration (Refs. 26-28). In the revised manuscript, we have further expanded this discussion and added additional references, which solidify the foundational role of QACs in halogen sequestration. However, current research predominantly focuses on the cationic moiety of complexing agents, where solid complexes formed between quaternary ammonium cations and polyhalide anions suppress the shuttle effect of polyhalides, thereby enhancing the Coulombic efficiency and extending the cycle lifespan of batteries. Nevertheless, such liquid-solid phase transformation strategies often exhibit sluggish reaction kinetics, which adversely impacts voltage efficiency and electrochemical reversibility, consequently compromising the overall energy efficiency of the battery system. To address these limitations, there remains an urgent need to develop advanced complexing agents that not only immobilize polyhalide anions to mitigate shuttle effects but also facilitate enhanced reaction kinetics, ultimately improving the holistic performance of energy storage systems. In this study, we employ a cation-driven phase transition approach to mitigate the shuttle effect and introduce an innovative anion-enhancement strategy to enhance reaction kinetics. We selected the simple tetramethylammonium cation (TMA^+) as the cationic component of the complexing agent and focused on investigating the impact and underlying mechanisms of different halide anions on the reaction kinetics. Notably, we innovatively proposed that F^- , as the anionic component of the complexing agent, significantly enhances the reaction kinetics. In the TMAF-modified electrolyte, the zinc-interhalide complex battery (ZICB) demonstrates an exceptionally high average energy efficiency (AEE=95.2%) and excellent reversibility.

We also sincerely appreciate the reviewer for highlighting the emerging role of QACs as surface adsorbents in zinc-ion batteries. In a study by Zhang et al.,

tetrabutylammonium p-toluenesulfonate was used as an additive to modulate the solvation structure of Zn^{2+} through anion regulation. The cation TBA⁺ preferentially adsorbs at the tips of the Zn electrode surface, thereby inhibiting dendrite growth and suppressing side reactions (Nat. Commun. 2023, 14, 3526). Similarly, Liang et al. found that TPA⁺ preferentially adsorbs at the tips, effectively blocking the strong electric field and regulating the ion distribution at the interface, which helps to suppress dendrite growth (iScience 2020, 23, 101348). However, these studies have primarily focused on the role of QACs in inhibiting dendrite growth, lacking comprehensive characterization of the zinc anode interface, and have not explored the advantageous crystal facets induced by QACs, nor provided computational insights into the underlying mechanisms. In this study, we performed an extensive characterization of the zinc anode surface to elucidate the dendrite-suppressing mechanism. A series of high-precision techniques, including Electron Backscatter Diffraction (EBSD) and 2-dimensional Grazing-Incidence Wide-Angle X-ray Scattering (2D-GIWAXS), confirmed that TMA⁺ promotes the transition of zinc deposition from disordered growth to ordered deposition on the (101) crystal facet. Furthermore, density functional theory (DFT) calculations were conducted to compare the adsorption energies of TMA⁺, H₂O, and Zn²⁺ on different Zn crystal facets. Compared to the Zn (101) plane, TMA⁺ exhibits lower adsorption energy on the Zn (002) and Zn (100) planes, making it more inclined to adsorb on these planes. This nontraditional exposure of Zn (002) and Zn (100) surfaces favors their growth, thereby leading to the development of a Zn (101)-dominant texture. The revised manuscript now includes 3 additional references (Refs. 29, 30, 53) covering both foundational and cutting-edge QAC studies. We believe these additions better position our work within the historical context while highlighting its unique contributions. In summary, we make innovations from three aspects in this work: **ultra-high energy efficiency**, **comprehensive study on dual electrode-electrolyte interface** and **innovative emphasis on fluorine (F) chemistry**.

1. Ultra-high energy efficiency. We introduce a cation-driven cathode phase transition to suppress the shuttle effect and achieve uniform zinc electrodeposition, along with a synergistic anion kinetic enhancement strategy to enhance the battery's EE and lifespan. TMA⁺ cation promotes the oriented deposition of Zn on the anode through electrostatic shielding, significantly extending cycling life. Concurrently, it captures I₃⁻ on the cathode, forming a stable solid-phase interhalide complex that enhances coulombic efficiency (CE). Compared to I₃⁻, the introduction of X⁻ anions lowers the Gibbs free energy differences (ΔG) of I⁻→I₂X⁻ and I₂X⁻→TMAI₂X and accelerates I⁻/I₂X⁻/TMAI₂X conversions, thereby improving voltage efficiency (VE). In TMAF-modified electrolytes, zinc interhalide complex batteries (ZICBs) achieve an ultra-high EE of 95.2% at 0.2 A g⁻¹. At 1 A g⁻¹, they show an exceptionally low decay rate of 0.01‰ per cycle across 10,000 cycles.

2. Comprehensive study on dual electrode-electrolyte interface. We discovered that the electrode interfaces can undergo a complete transformation through the designed strategy. On the anode side, zinc deposition transitions from disordered deposition to an ordered stacking on the (101) crystal plane, as confirmed by a series of high-precision characterizations, such as Electron backscatter diffraction (EBSD), 2-dimensional grazing-incidence wide-angle X-ray scattering (2D-GIWAXS), and others. Meanwhile, we unveiled the underlying mechanism behind this interfacial electrochemical behavior reversal. On the cathode side, the interface shifts from the conventional liquid-liquid interface to a liquid-solid electrochemical behavior, fundamentally addressing the issues associated with polyhalide derivatives.

3. Innovative emphasis on F chemistry. We have investigated the effect of halogen anions on the thermodynamics and kinetics of the I⁻/I₂X⁻ conversion and emphasized the importance of F chemistry in achieving high EE and prolonging the lifespan of Zn-halogen batteries for the first time. On the one hand, I⁻ and X⁻ combine undergo electron loss to form I₂X⁻, where X⁻ lowers the reaction's Gibbs free energy differences (ΔG , with F having the most

pronounced effect), making the redox process energetically favorable. On the other hand, TMA^+ combines with I_2X^- to form solid TMAI_2X . The $\text{I}_2\text{F}^- \rightarrow \text{TMAI}_2\text{F}$ reaction has the lowest ΔG , making it the most easily formed. Therefore, F chemistry plays a dual role in enhancing the entire reaction pathway, contributing to improved reaction kinetics.

Thus, this work proposes new insights and principles in guiding electrolyte design for Zn-halogen batteries.

29. Lim, Y., Lee, G., Kim, J.H., Seo, J.K. & Yoo, S.J. Tetrabutylammonium bromide incorporated hydrated deep eutectic solvents: Simultaneously addressing anode stability and cathode efficiency in zinc-bromine batteries. *Energy Storage Mater.* **68**, 103331 (2024).

30. Gao, L. et al. A High-Performance Aqueous Zinc-Bromine Static Battery. *iScience* **23**, 101348 (2020).

53. Chen, S. et al. Coordination modulation of hydrated zinc ions to enhance redox reversibility of zinc batteries. *Nat. Commun.* **14**, 3526 (2023).

Q2. "...this liquid-solid conversion often exhibits slow reaction kinetics, impacting reversibility." According to this draft, TMAI_2F is the most thermodynamically stable, demonstrating the exceptional ability of TMAF to capture polyiodine. However, how do these results indicate that TMAF effectively boosts the kinetic performance of both oxidation and reduction processes, especially in battery applications related to discharge power output? How does the solid complex formation correlate with the electrochemical reversibility?

Response: We thank the reviewer for this valuable and constructive question. The $\text{I}^- \rightarrow \text{TMAI}_2\text{X}$ reaction primarily involve two steps (Fig. 3k): **Step I: $2\text{I}^- - 2\text{e}^- + \text{X}^- \rightarrow \text{I}_2\text{X}^-$** , I^- and X^- combine undergo electron loss to form I_2X^- ; **Step II: $\text{TMA}^+ + \text{I}_2\text{X}^- \rightarrow \text{TMAI}_2\text{X}$** , TMA^+ combines with I_2X^- to form solid TMAI_2X . To investigate the impact of TMAX on redox kinetics and examine the relationship

between the formation of the TMAI_2X solid complex and electrochemical reversibility, density functional theory (DFT) was employed to calculate the Gibbs free energy differences (ΔG) for step I and step II, with the results presented in Fig. 3g and Supplementary Table S3. The ΔG values for both step I and step II follow the trend $\text{F} < \text{Cl} < \text{Br} < \text{I}$, with the ΔG for step I being lower than that for step II. A lower ΔG typically corresponds to a stronger thermodynamic driving force, which favors an increased reaction rate and better kinetic performance (Nat. Commun. 2023, 14, 1856; Energy Environ. Sci., 2023, 16, 4630-4640; Energy Environ. Sci., 2023, 16, 4073–4083; Adv. Funct. Mater. 2024, 2422868; Nano Energy 2025, 133, 110519; Adv. Sci. 2024, 11, 2410653).

The improvement in kinetic performance by TMAX is achieved through a two-step mechanism (Fig. 3k). In step I, I^- and X^- combine and undergo electron loss to form I_2X^- , where X^- reduces the reaction's ΔG (with F having the most pronounced effect), making the redox process thermodynamically favorable and accelerating the reaction rate. In step II, TMA^+ combines with I_2X^- to form solid TMAI_2X , where the ΔG of F is the lowest, providing a stronger driving force that enhances the kinetic performance. In Step II, TMA^+ combines with I_2X^- to form the solid TMAI_2X , where the ΔG value for F is the lowest, providing a stronger driving force. This makes TMAI_2F more easily formed, promoting the conversion of $\text{I}_2\text{X}^- \rightarrow \text{TMAI}_2\text{X}$, enhancing the reversibility of the reaction and improving the kinetic performance.

Therefore, TMAF can lower the ΔG of both step I and step II, playing a crucial role in enhancing redox reaction kinetics (step I) and promoting the formation of solid complexes, thereby improving electrochemical reversibility (step II).

Detailed information is provided in the Results and discussion section of the manuscript.

Halide Anion-Enhanced Cathode Reaction Kinetics

To explain the kinetic differences among halide ions, density functional theory

(DFT) calculations were performed. Gibbs free energy differences (ΔG) for reactions of $I^- \rightarrow I_2X^- \rightarrow TMAI_2X$ ($X = F, Cl, Br, I$) were calculated (Fig. 3g and Supplementary Table S3). The ΔG values for $I^- \rightarrow I_2X^-$ follow the trend $I_2F^- < I_2Cl^- < I_2Br^- < I_3^-$. Lower ΔG values indicate higher reaction spontaneity, with I_2F^- being the most favorable and stable due to its lowest ΔG . This suggests that TMAF facilitates the acceleration of the reaction rate and enhances the iodide redox kinetics.⁴⁶⁻⁴⁸ Structural analysis (Fig. 3h) revealed that all I_2X^- species exhibit linear or near-linear trihalide structures.⁴⁹ Among them, I_2F^- has the shortest I–X bond length, indicating the strongest binding affinity, which decreases in the order $F > Cl > Br > I$. Electrostatic potential (ESP) distributions (Fig. 3i and Supplementary Fig. S27) show that iodine atoms in I_3^- carry negative charges, while bonding with electronegative halogens (F, Cl, Br) causes electron transfer, resulting in the I atom adjacent to X^- carrying a partial positive charge. F, with its high electronegativity, induces the highest positive charge on the adjacent I atom. These findings are consistent with XPS observations for the cathode electrode (Figs. 2g-i and Supplementary Figs. S14, S15).

The ΔG values for the reaction $I_2X^- \rightarrow TMAI_2X$ also follow the trend: $TMAI_2F < TMAI_2Cl < TMAI_2Br < TMAI_3$ (Fig. 3g). Lower ΔG values favor the reaction, indicating that $TMAI_2F$ is more easily formed and exhibits superior kinetic performance. We also calculated the adsorption energy of $TMAI_2X$ on the activated carbon electrode surface (Fig. 3j and Supplementary Figs. S28-31). The adsorption energy of $TMAI_2X$ on the activated carbon electrode follows the trend: $TMAI_2F < TMAI_2Cl < TMAI_2Br < TMAI_3$. Among them, the adsorption energy of $TMAI_2F$ was the lowest, which indicated that it had the strongest adsorption force and was most likely to be attached to the surface of the activated carbon electrode. This stronger adsorption energy implies that $TMAI_2F$ will firmly adhere to the electrode surface and diffuse more rapidly, leading to a more uniform distribution. In contrast, the adsorption energies of the $TMAI_2Cl$ and $TMAI_2Br$ complexes are weaker, which may result in larger adsorption-

desorption fluctuations, leading to unstable adsorption on the electrode surface and a more uneven distribution. This is consistent with experimental observations (Figs. 2c, f).

The computational results align with the aforementioned experimental findings, demonstrating that TMAF significantly enhances the reaction kinetics and ensures uniform distribution on the activated carbon electrode surface. The enhancement of the $I^- \rightarrow TMAI_2X$ reaction occurs through a two-step mechanism (Fig. 3k). In Step I (Eq. 3), I^- and X^- combine and undergo electron loss to form I_2X^- , where X^- lowers the reaction's ΔG (with F having the most pronounced effect), making the redox process energetically favorable. In Step II (Eq. 4), TMA^+ combines with I_2X^- to form solid $TMAI_2X$. The $I_2F^- \rightarrow TMAI_2F$ reaction has the lowest ΔG , making it the most easily formed. Therefore, F chemistry plays a dual role in enhancing the entire reaction pathway, contributing to improved reaction kinetics.

Fig. 3| Kinetic analysis of ZICBs using TMAX electrolyte. **a** CV profiles of ZICBs using TMAF electrolyte. **b** Relationship between peak current and square root of scan rate. Tafel plots of **c** IOR and **d** IRR. **e** I⁻ diffusion coefficient in different electrolytes. **f** DRT curves charging to different voltage. **g** The Gibbs free energy ladder diagram. **h** Configurational structure and bond length of I₂X⁻. **i** Valence states (atomic Mulliken charge) and ESP of I₂F⁻. **j** Adsorption energy TMAI₂X on carbon layer. **k** Schematic illustration of the halide anion-enhanced behavior.

Supplementary Table S3. Gibbs free energies of I₂X⁻ and TMAI₂X and Gibbs free energy differences (ΔG) of step I and step II. (Unit: eV)

X	I ₂ X ⁻	TMAI ₂ X	Step I (ΔG)	Step II (ΔG)
F	-4.11	-6.57	-4.11	-2.46
Cl	-3.37	-4.74	-3.37	-1.37
Br	-3.12	-4.27	-3.12	-1.15
I	-2.89	-3.84	-2.89	-0.95

Q3. Please provide experimental details on how to chemically synthesize the KI_3 solution exclusively and how to characterize its formation.

Response: We thank the reviewer for this valuable question. KI_3 was generated using the following chemical reactions: $\text{KI} + \text{I}_2 \rightarrow \text{KI}_3$. (Adv. Energy Mater. 2023, 2302187; Energy Environ. Sci., 2017, 10, 735—741; J. Am. Chem. Soc. 2016, 138, 9373–9376) Based on this reaction, KI_3 can be prepared by mixing excess KI and I_2 . The specific experimental details are as follows: add 1.6600 g of KI to 10 mL of deionized water and stir until fully dissolved. Then, introduce 0.1524 g of I_2 into the resulting solution and stir for approximately 10 minutes to obtain the KI_3 solution. The resulting KI_3 solution is dark orange (Supplementary Fig. S2). Detailed information is provided in the Methods section.

Preparation of KI_3 solution

Add 1.6600 g of KI to 10 mL of deionized water and stir until fully dissolved. Then, introduce 0.1524 g of I_2 into the resulting solution and stir for approximately 10 minutes.

Supplementary Fig. S2. Photograph of KI_3 solution.

To determine the formation of KI_3 , the prepared solution was analyzed using Raman and UV-Vis spectroscopy. In the Raman spectrum (Supplementary Fig. S3), the peaks at 118 and 136 cm^{-1} are attributed to I_3^- , while the peak at 165

cm^{-1} corresponds to I_5^- (Energy Storage Mater. 2023, 54, 339-365; Adv. Funct. Mater. 2022, 32, 2111026). The formation of I_5^- is likely a result of the disproportionation reaction: $2\text{I}_3^- \rightarrow \text{I}_5^- + \text{I}^-$. Additionally, the UV-Vis spectrum (Supplementary Fig. S4) reveals characteristic absorption peaks of I_3^- at 288 nm and 350 nm (Adv. Energy Mater. 2024, 14, 2400110). These findings collectively confirm the formation of KI_3 in the solution.

Supplementary Fig. S3. Fitted Raman spectra of KI_3 solution.

Supplementary Fig. S4. UV-vis spectra of KI_3 solution.

Q4. The detailed information on activated carbon (e.g., surface area, pore

volume) is missing. Is there any correlation between the size of the complex and the pores of the activated carbon in retaining charged products? Does TMAX become confined within the pores, or does it simply cover the surface?

Response: We thank the reviewer for this valuable comment. In this study, commercial activated carbon (YP80F) was used, which has a specific surface area of $2152 \text{ m}^2 \text{ g}^{-1}$ and a pore volume of $0.94 \text{ cm}^3 \text{ g}^{-1}$. To investigate the distribution of the charging product TMAI_2X on the activated carbon electrode, we first conducted nitrogen (N_2) adsorption experiments at 77 K on the uncharged activated carbon electrode (excluding the carbon paper part) and on activated carbon electrodes charged to 1.6 V in ZnSO_4+KI and TMAF electrolytes, respectively. These experiments were performed to characterize the specific surface area (S_{BET}) and pore volume of the activated carbon before and after charging in different electrolytes (Supplementary Fig. S20 and Supplementary Table S1). S_{BET} and pore volume follow the order: commercial activated carbon powder > uncharged activated carbon > activated carbon electrodes charged to 1.6 V in TMAF electrolyte > activated carbon electrodes charged to 1.6 V in ZnSO_4+KI electrolyte. When the activated carbon powder and PVDF are mixed and coated to form the electrode, PVDF acts as a binder and may block or fill some of the pores, leading to a significant reduction in both the S_{BET} and pore volume. After charging the activated carbon electrodes to 1.6 V, both the S_{BET} and pore volume decrease compared to the uncharged electrodes, regardless of whether the electrolyte is ZnSO_4+KI or TMAF, due to the formation of charging products. For the activated carbon in ZnSO_4+KI electrolyte, I_2 is generated during charging, which reacts with I^- in the electrolyte to form I_3^- . The dissolution of the active species leads to a larger decrease in S_{BET} and pore volume, compared to the activated carbon in TMAF electrolyte, where stable TMAI_2F is formed. The decrease in S_{BET} and pore volume values after charging indicates that the charging product of TMAF, TMAI_2F , was able to be confined in the pores of the activated carbon.

Supplementary Fig. S20. N₂ porosimetry measurements for **a** uncharged and activated carbon electrodes charged to 1.6 V in **b** ZnSO₄+KI and **c** TMAF electrolytes, respectively.

Supplementary Table S1. S_{BET} and pore volume of activated carbon before and after charging in different electrolytes.

Type of activated carbon	S _{BET} (m ² g ⁻¹)	Pore volume (cm ³ g ⁻¹)
Commercial activated carbon powder	2152	0.94
Uncharged	509	0.27
Charged to 1.6 V in ZnSO ₄ +KI electrolyte	400	0.21
Charged to 1.6 V in TMAF electrolyte	379	0.20

Based on the SEM images of the positive electrode after charging in Fig. 2a, it can be seen that small flakes were formed on the cathode surface in both ZnSO₄+KI and TMAF electrolytes. However, in the ZnSO₄+KI electrolyte, the distribution of these flakes on the electrode surface is not regular. In contrast, in the TMAF electrolyte, the flakes were more uniformly and tightly attached to the surface of the bulk activated carbon particles.

To visually observe the distribution of charging products on the surface and within the pores of the activated carbon particles, we employed Focused Ion Beam-Scanning Electron Microscope (FIB-SEM) to investigate the cross-section of a single activated carbon particle in ZnSO₄+KI and TMAF electrolytes. Elemental distribution in the cross-section was analyzed using EDS mapping

and line scanning (Supplementary Figs. S20, S21). In the TMAF electrolyte, the EDS mapping in Supplementary Fig. S20c reveals that although both I and F elements are distributed across the entire cross-section of the activated carbon particle, these elements are primarily concentrated in the gaps on the surface of the activated carbon and at the opposite surface of the particle. The EDS line scanning results in Supplementary Fig. S20e further confirm that I and F signals are observed along the line across the entire cross-section, with prominent signal peaks appearing at approximately 0.4 μm and 4 μm from the two surfaces of the activated carbon. In the ZnSO_4+KI electrolyte, the EDS mapping in Supplementary Fig. S21c shows that I is present across the entire cross-section of the activated carbon particle, but the signal intensity of I is significantly weaker compared to that in the TMAF electrolyte. Moreover, there is no evident I accumulation on the surface of the activated carbon. The EDS line scanning in Supplementary Fig. S20e also shows that the I signal on the surface is only slightly higher than that within the interior, which can be attributed to the dissolution of the charging product I_2 and the formation of the soluble I_3^- species.

These results indicate that the stable charging product TMAI_2X , formed in TMAF electrolytes, is distributed both on the surface and within the pores of the activated carbon, with the majority of the distribution occurring on the surface of the activated carbon. This conclusion has been supplemented in the Results and discussion section of the manuscript.

Supplementary Fig. S21. **a** Cross-section images of activated carbon after charging to 1.6 V in TMAF electrolyte. **b** Magnified images of **a**. **c** EDS element mapping. **d**, **e** EDS line scanning.

Supplementary Fig. S22. **a** Cross-section images of activated carbon after charging to 1.6 V in ZnSO₄+KI electrolyte. **b** Magnified images of **a**. **c** EDS element mapping. **d**, **e** EDS line scanning.

Q5. According to the manuscript, the polyiodide ion capture efficacy follows the order: TMAF > TMACl > TMABr. What is the main reason that among I₂X⁻ species, I₂F⁻ exhibits a more uniform distribution on the activated carbon electrode?

Response: We thank the reviewer for this constructive question. In the I₂X⁻ species, I₂F⁻ exhibits a more uniform distribution on the activated carbon electrode, which can primarily be attributed to the adsorption characteristics of the TMAI₂F complex on the electrode surface.

Specifically, it is known that TMAX (X = F, Cl, Br) reacts with polyiodide ions to form TMAI₂X complexes. To explain the distribution of TMAI₂X on the activated carbon electrode, DFT calculations were performed to determine the adsorption energy of TMAI₂X on the electrode surface, with the results and models shown

in Fig. 3j and Supplementary Figs. S28-31. Fig. 3j demonstrates that the adsorption energy of TMAI_2X on the activated carbon electrode follows the trend: $\text{TMAI}_2\text{F} < \text{TMAI}_2\text{Cl} < \text{TMAI}_2\text{Br} < \text{TMAI}_3$. The TMAI_2F complex exhibits the most negative adsorption energy, indicating the strongest adsorption force and a higher tendency to adsorb onto the activated carbon electrode surface. This stronger adsorption energy means that TMAI_2F will be firmly attached to the electrode surface and can diffuse more quickly, leading to a more uniform distribution. In contrast, the adsorption energies of TMAI_2Cl and TMAI_2Br complexes are weaker, which may lead to larger adsorption-desorption fluctuations, resulting in less stable adsorption on the electrode surface and a more uneven distribution.

In summary, the more uniform distribution of I_2F^- on the activated carbon electrode is primarily due to the strongest adsorption of the TMAI_2F complex on the electrode surface, ensuring a more stable and uniform distribution of I_2F^- across the electrode. This explanation has been supplemented in the Results and discussion section of the manuscript.

Fig. 3| j Adsorption energy TMAI_2X on carbon layer.

Supplementary Fig. S28. Adsorption of TMAI₂F on carbon layer.

Supplementary Fig. S29. Adsorption of TMAI₂Cl on carbon layer.

Supplementary Fig. S30. Adsorption of TMAI₂Br on carbon layer.

Supplementary Fig. S31. Adsorption of TMAI₃ on carbon layer.

Q6. "...These comprehensive analyses confirm the formation and uniform distribution of interhalide complexes TMAI₂X on the electrode, highlighting TMAX's role in stabilizing the electrode surface." What is meant by "stabilizing"? Isn't this simply deposition?

Response: We sincerely appreciate the reviewer's insightful question regarding the term "stabilizing" in our discussion. In zinc-iodine batteries, the stability of the iodine cathode is primarily influenced by iodine dissolution and the polyiodide shuttle effect. In the traditional KI electrolyte before modification, I₂ is formed during charging, and the strong interaction between I₂ and I⁻ often leads to liquid-liquid conversion between I⁻/I₃⁻, which may result in the shuttle effect, causing a decrease in coulombic efficiency and capacity fading. However, after modification, the formation of solid-phase TMAI₂X halide complexes at the electrode surface suppresses iodine dissolution (i.e., the formation of I₃⁻), leading to a more stable electrode interface compared to the unmodified system. We use the term "**stabilizing**" rather than "**deposition**" here to specifically emphasize the role of TMAX in inhibiting the dissolution of iodine and the formation of polyiodide anions at the cathode.

7. Solid bromine complexes are not formed with short-alkyl-chain tetramethyl and tetraethyl ammonium salts. Why, then, is a solid complex formed with TMA⁺ in this case?

Response: We thank the reviewer for this valuable question. The formation of solid complexes between quaternary ammonium cations and polyhalide anions can be analyzed based on the strength of their interactions.

Studies have shown that the longer the alkyl chain of the quaternary ammonium cation, the stronger the electrostatic interaction between the quaternary ammonium cation and polybromide anion, which effectively captures Br₂ and prevents its crossover (Nano Lett. 2024, 24, 13796–13804). A similar conclusion has been drawn regarding the interaction between quaternary ammonium cations and polyiodide anions (Chem. Sci. 2024, 15, 3357–3364). Indeed, TMA⁺ does not form a solid-state complex with polybromide anions; however, the formation of solid-state complexes between tetraethylammonium cations (TEA⁺) and polybromide anions has been demonstrated (iScience 2020, 23, 101348).

TMA⁺ can form solid complexes with I₃⁻, but not with Br₃⁻. This phenomenon can be explained by the Hard-Soft Acid-Base (HSAB) theory (J. Am. Chem. Soc. 2017, 139, 9985). TMA⁺ has a low charge density (with the positive charge dispersed over four methyl groups) and high polarizability, making it a soft acid or borderline acid. I₃⁻ has a large volume and high polarizability, making it a typical soft base, whereas Br₃⁻ has weaker polarizability and a smaller volume, closer to a borderline base. According to the HSAB theory, the soft-soft interaction between TMA⁺ (soft acid) and I₃⁻ (soft base) is stronger, resulting in a more stable complex. In contrast, the poor match between Br₃⁻ (borderline base) and TMA⁺ makes it difficult to form a solid complex.

8. Zn||Zn symmetric cells use Whatman GF/D as the separator, while Zn-interhalide complex batteries (ZICBs) employ Whatman GF/A. Is there a specific reason for this choice? Does it significantly affect cell performance?

Response: We thank the reviewer for this valuable question. Whatman GF/D (675 μm thick), compared to GF/A (260 μm thick), has a greater thickness, which can extend the time for zinc deposition through the separator and improve the battery's cycle life (J. Membr. Sci. 2023, 688, 122130; Energy Fuels 2022, 36, 8, 4609–4615). In ZnSO_4+KI electrolyte, $\text{Zn}||\text{Zn}$ symmetric cells with Whatman GF/A separator exhibit a very short lifespan (Fig. R8). To better compare the cycling performance of $\text{Zn}||\text{Zn}$ symmetric cells in ZnSO_4+KI electrolyte and TMAX electrolyte, we selected Whatman GF/D as the separator for the $\text{Zn}||\text{Zn}$ symmetric cells.

Fig. R8. Galvanostatic cycling stability of symmetric $\text{Zn}||\text{Zn}$ cells using ZnSO_4+KI electrolyte and Whatman GF/A separator under 4 mA cm^{-2} , 1 mAh cm^{-2} .

$\text{Zn}||\text{Zn}$ symmetric cells with Whatman GF/A separator in TMAF electrolyte also exhibit excellent performance. The cells using TMAX electrolytes, at a current density of 4 mA cm^{-2} and a capacity of 1 mAh cm^{-2} , maintained stable cycling for over 1200 hours (Figs. R9-10).

Fig. R9. Galvanostatic cycling stability of symmetric Zn||Zn cells using TMAF electrolyte and Whatman GF/A separator under 4 mA cm^{-2} , 1 mAh cm^{-2} .

Fig. R10. Enlarged views symmetric Zn||Zn cells in Fig. R9 of **a** 2-8 cycles, **b** 994-1000 cycles and **c** 2394-2400 cycles.

To explore the impact of TMAF on enhancing the stability of Zn electrodes under high area capacity conditions, we constructed symmetric cells for cycling at 5 mA cm^{-2} and 5 mAh cm^{-2} as outlined in Figs. R11-12. Cells with TMAF electrolytes maintained stable cycling for over 700 hours. The findings indicate that TMAF significantly improves the stability of Zn electrodes.

Fig. R11. Galvanostatic cycling stability of symmetric Zn||Zn cells using TMAF electrolyte and Whatman GF/A separator under 5 mA cm^{-2} , 5 mAh cm^{-2} .

Fig. R12. Enlarged views symmetric Zn||Zn cells in Fig. R11 of **a** 1-5 cycles, **b** 196-200 cycles and **c** 346-350 cycles.